# Abscisic acid dynamics in roots detected with genetically encoded FRET sensors

**Alexander M Jones[1]\*, Jonas ÅH Danielson[1], Shruti N ManojKumar[1], Viviane Lanquar[1], Guido Grossmann[1,2], Wolf B Frommer[1]\***

[1]Department of Plant Biology, Carnegie Institution for Science, Stanford, United States; [2]Centre for Organismal Studies, Cluster of Excellence CellNetworks, Universität Heidelberg, Heidelberg, Germany

**Abstract** Cytosolic hormone levels must be tightly controlled at the level of influx, efflux, synthesis, degradation and compartmentation. To determine ABA dynamics at the single cell level, FRET sensors (ABACUS) covering a range ~0.2–800 µM were engineered using structure-guided design and a high-throughput screening platform. When expressed in yeast, ABACUS1 detected concentrative ABA uptake mediated by the AIT1/NRT1.2 transporter. Arabidopsis roots expressing ABACUS1-2µ ($K_d$~2 µM) and ABACUS1-80µ ($K_d$~80 µM) respond to perfusion with ABA in a concentration-dependent manner. The properties of the observed ABA accumulation in roots appear incompatible with the activity of known ABA transporters (AIT1, ABCG40). ABACUS reveals effects of external ABA on homeostasis, that is, ABA-triggered induction of ABA degradation, modification, or compartmentation. ABACUS can be used to study ABA responses in mutants and quantitatively monitor ABA translocation and regulation, and identify missing components. The sensor screening platform promises to enable rapid fine-tuning of the ABA sensors and engineering of plant and animal hormone sensors to advance our understanding of hormone signaling.

**\*For correspondence:** ajones@carnegiescience.edu (AMJ); wfrommer@carnegiescience.edu (WBF)

**Competing interests:** The authors declare that no competing interests exist.

**Reviewing editor**: Richard Amasino, University of Wisconsin, United States

## Introduction

Research by the Intergovernmental Panel on Climate Change (IPCC) indicates that the climate is becoming more variable and that the frequency of floods and droughts is increasing (IPCC, 2013). Elevated temperatures combined with periods of drought represent a major threat to food security (*Sreenivasulu et al., 2007*; *Barnabas et al., 2008*; *Brutnell and Frommer, 2012*; *Sreenivasulu et al., 2012*). Thus, there is an obvious and urgent need to advance our understanding of plant tolerance to these stresses as well as the underlying mechanisms as a basis for engineering crops that can survive the anticipated environmental challenges (*Tester and Langridge, 2010*; *Schroeder et al., 2013*).

Abscisic acid (ABA), a terpenoid plant hormone, has been found in all kingdoms of life, including plant saprophytic and pathogenic fungi, animals such as sponges (*Axinella polypoides*) and hydroids (*Eudendrium racemosum*), and human parasites (*Toxoplasma gondii*) (*Crocoll et al., 1991*; *Nagamune et al., 2008*; *Li et al., 2011*). Yet, we know little regarding the homeostasis and dynamics of this hormone in these systems. In plants, abscisic acid (ABA) serves as the key phytohormone produced during drought, and is a master regulator of water use efficiency, stomatal aperture, and other mechanisms of tolerance to drought and osmotic stress (*Sreenivasulu et al., 2012*). ABA also controls seed dormancy and germination (*Finkelstein and Rock, 2002*; *Bentsink and Koornneef, 2008*; *Kanno et al., 2010*).

ABA is perceived by a family of ABA receptors (PYR/PYL/RCAR proteins, *Ma et al., 2009*; *Park et al., 2009*), which then bind to and prevent ABA co-receptor proteins (protein phosphatase 2CA proteins, PP2CAs, *Umezawa et al., 2009*; *Vlad et al., 2009*) from deactivating Sucrose non-fermenting Related Kinase 2s (SnRK2s, *Fujii and Zhu, 2009*; *Fujita et al., 2009*; *Nakashima et al., 2009*). The

**eLife digest** Plants are able to respond to detrimental changes in their environment—when, for example, water becomes scarce or the soil becomes too salty—in ways that minimize stress and damage caused by these changes. Hormones are chemicals that trigger the plant's response under these circumstances.

Abscisic acid is the hormone that regulates how plants respond to drought and salt stress, and also controls growth and development. In the past, it was possible to measure the average level of this hormone in a given tissue, but not the level in individual cells in a living plant, nor in specific compartments within a cell. Moreover, it was difficult to follow directly how abscisic acid moved between the plant cells, tissues or organs.

Now, Jones et al. (and independently Waadt et al.) have developed tools that can measure the levels of abscisic acid within defined compartments of individual cells in living plants and in real time. The plants were genetically engineered to produce sensor proteins with two properties: they can bind to abscisic acid in a reversible manner, and they contain two 'reporters' that fluoresce at different wavelengths. Shining light onto the plant at a specific wavelength that is only absorbed by one of the reporters causes both of the reporters on the sensor proteins to fluoresce. However, the two reporters fluoresce differently when the sensor binds to abscisic acid. Specifically, one reporter fluoresces more and the other less. Hence, measuring the ratio of these two wavelengths in the light that is given off by the sensor proteins can be used as a measure of the concentration of abscisic acid in a plant cell.

Jones et al. used a high-throughput platform to engineer five sensor proteins that detect abscisic acid over a wide range of concentrations. Using these 'ABACUS' sensors in living plants could track the uptake of abscisic acid into root cells, and revealed that the concentration of the hormone inside the cell stayed below the levels provided on the outside. Since known abscisic acid-transporters are capable of raising the hormone concentration inside a cell above that provided on the outside, abscisic acid transport into plant roots may occur via as-yet-undiscovered transporter proteins. Jones et al. also show that root cells rapidly eliminate abscisic acid, and that adding extra abscisic acid to the roots increases the rate of elimination within minutes.

Plants were also engineered to target the sensor proteins specifically to the cell nucleus. In the future, targeting these sensors to the cell wall should allow tracking of the cell-to-cell movement of this hormone. Further aims include using ABACUS to track abscisic acid in plants undergoing stress, and to use the high-throughput platform to develop new sensors to track other hormones in living organisms (including animals).

resulting increase in SnRK2 activity triggers phosphorylation of transcription factors (*Yoshida et al., 2010*) and transporters (*Geiger et al., 2009*; *Lee et al., 2009*; *Sato et al., 2009*), and affects ABA-dependent responses that vary with respect to cell type and developmental or environmental context. For example, only ~25% of the ABA responsive transcripts in leaves are also ABA responsive in guard cells (*Wang et al., 2011*), and ABA signaling specifically in the endodermis promotes lateral root quiescence in plants exposed to salt stress (*Duan et al., 2013*). ABA responses affect many aspects of physiology and development, evidenced by pleiotropic phenotypes of mutants deficient in ABA metabolism (*Leon-Kloosterziel et al., 1996*), perception (*Gonzalez-Guzman et al., 2012*), or signaling (*Rubio et al., 2009*; *Fujii et al., 2011*). Hallmark responses include stomatal closure (*Leung and Giraudat, 1998*), and altered growth rates and metabolic adjustments during stress responses (*Ober and Sharp, 1994*; *Sharp and LeNoble, 2002*), senescence (*Hunter et al., 2004*) and seed dormancy (*Bentsink and Koornneef, 2008*; *Rodriguez-Gacio Mdel et al., 2009*). Additionally, ABA affects other processes, including immunity (*Cao et al., 2011*), crosstalk signaling with sugars (e.g., *Arenas-Huertero et al., 2000*; *Matiolli et al., 2011*) and other plant hormones (*Jaillais and Chory, 2010*). Thus, simple tuning of ABA levels broadly in order to change drought tolerance carries side effects such as increased seed dormancy (e.g., *Lin et al., 2007*).

Understanding the diverse roles of ABA in multiple tissues requires quantitative knowledge of ABA levels and dynamics in individual cells. ABA accumulation under stress conditions is controlled at the level of biosynthesis, catabolism, and transport as well as through reversible conversion to

ABA-glucose ester, an inactive form of ABA potentially used in both storage and long distance transport (*Cutler and Krochko, 1999*; *Seo and Koshiba, 2002*; *Wilkinson and Davies, 2002*; *Nambara and Marion-Poll, 2005*). Many of the enzymes involved in ABA metabolism have been identified, some of which are present as multi-gene families (*Tan et al., 2003*; *Kushiro et al., 2004*; *Lee et al., 2006*; *Xu et al., 2012*).

The ubiquitous distribution of biosynthetic enzymes and receptors for hormones has raised the question whether they function locally or remotely from the site of synthesis; remote action is a hallmark of animal hormones. Long distance transport of auxin is critical (*Grieneisen et al., 2007*), yet there has been a long-standing debate on whether and how hormones such as ABA are transported between plant organs. The weak acid properties of ABA (pK$_a$ 4.7) led to the hypothesis that ABA diffuses freely across membranes in its undissociated lipophilic form (*Kaiser and Hartung, 1981*). Subsequent studies implicated cellular uptake via transport proteins (*Astle and Rubery, 1985b*). Over the past few years, functional studies identified proteins that can mediate transport of ABA (*Kang et al., 2010*; *Kuromori et al., 2010*, *2011*; *Kanno et al., 2012*).

ABA levels can be monitored indirectly by expressing GUS, GFP or luciferase reporters under the control of ABA responsive promoters (*Christmann et al., 2005*; *Kim et al., 2011*; *Duan et al., 2013*; *Geng et al., 2013*), but the indirect nature of ABA detection of these reporters results in several potential limitations. For example, promoters may not respond in a linear fashion and can be subject to additional regulatory inputs. Furthermore, such reporters cannot detect rapid changes (seconds to minutes) and are susceptible to both variation in the activities of ABA signaling components and nonspecific regulation of expression levels resulting from crosstalk with other signaling pathways (common in hormone regulated gene expression) (*Jaillais and Chory, 2010*). Improved direct measurement of ABA through biochemical methods such as in situ mass spectrometry (*Lorenzo Tejedor et al., 2012*) could increase the spatial resolution of ABA quantitation, but does not allow for dynamic measurement in live plants. Thus, direct sensors for ABA would represent a significant step forward in allowing the investigation of ABA levels with high spatial and temporal resolution. One of the most advanced technologies for high-resolution measurement of small molecules in living tissues is based on genetically encoded, ratiometric fluorescent sensors that bind to and report on the levels of the target molecule (i.e., sensors based on Förster Resonance Energy Transfer; FRET sensors [*Okumoto et al., 2012*]). FRET sensors are fusion proteins that report target molecule interactions through changes in the conformation of intrinsic sensory domains. These conformational changes affect the efficiency of energy transfer from a fused FRET donor fluorescent protein to a fused FRET acceptor fluorescent protein. Changes in energy transfer can be detected by measuring changes in the relative intensity of the two fluorescent proteins (ratio change) after excitation of the donor; the ratio change reports target molecule concentration. FRET sensors have been used in plant tissues to study calcium and zinc dynamics with subcellular spatial and near real-time temporal resolution (e.g., *Krebs et al., 2012*; *Lanquar et al., 2014*). Metabolite dynamics have also been studied using FRET sensors leading to, for example, the characterization of sugar transport in roots of intact seedlings (*Deuschle et al., 2006*; *Chaudhuri et al., 2008*, *2011*; *Grossmann et al., 2011*; *Grossmann et al., 2012*) and the identification of novel sugar transporters (*Chen et al., 2010*, *2012*; *Xuan et al., 2013*).

Here we developed a combinatorial and iterative engineering platform for FRET sensors that led to the identification of ABACUS1, an A̲b̲scisic A̲cid Concentration and U̲ptake S̲ensor version 1 that reports ABA levels. The engineering platform is based on a series of Gateway destination vectors coding for FRET pairs and a library of Entry clones coding for potential ABA sensory domains. Destination vector series for expression of potential sensor fusions in bacteria, yeast or plants were generated. The combinatorial screening of the Destination vector series and the Entry clone library allowed for rapid testing of hundreds of constructs and led to the identification of ABACUS1, which is specific for ABA and is optimized for reduced size, fluorophore brightness and photostability, and high ratio change (i.e. high signal-to-noise). We then generated ABACUS1 variants covering a broad detection range for in vivo use. We studied the properties of a known ABA transporter through analysis of yeast cells coexpressing ABACUS1. Using ABACUS1 sensors in the cytosol and nucleus of Arabidopsis, we detected reversible and concentration-dependent accumulation of exogenous ABA delivered to roots growing in the RootChip (*Grossmann et al., 2011*, *2012*), a microfluidic device for continuous measurement of living roots in tightly controlled environmental conditions. ABA feedback on ABA levels was identified as a potentially critical aspect of ABA signaling as time-course analysis of ABACUS responses demonstrated negative regulation of cytosolic ABA levels by ABA.

## Results

### ABA FRET sensor engineering

Generation of high sensitivity FRET sensors is inherently empirical and requires multi-parameter optimization (affinity, brightness, dynamic range, etc) (*Okumoto, 2012*). Selection of the sensory domain is a critical step for designing FRET sensors, as it determines affinity and specificity. To generate FRET sensors for ABA, potential ABA sensory domains (PAS) were selected from members of ABA co-receptor complexes; specifically, nine members of the PYR/PYL/RCAR family of ABA receptors and three members of the PP2CA sub-family of ABA co-receptor phosphatases from *Arabidopsis* (*Table 1*, *Supplementary file 1*). We generated two types of sensors: single domain sensors in which either a PYR/PYL/RCAR or PP2CA polypeptide alone (sPAS) is sandwiched between two fluorescent proteins, or double domain sensors, in which a PYL/PYR/RCAR fused via a linker to a PP2CA (dPAS) is sandwiched between two fluorescent proteins (*Figure 1A*). To cover a large test space of ABA sensory domains, we engineered sensor constructs using >50 sPAS and dPAS variants sandwiched between a fluorescent protein FRET pair (i.e., multiple PAS Entry clones were recombined with a single destination vector (*Figure 1—figure supplement 1*).

In parallel, we created an array of sensor constructs using a subset of the PAS domains combined with a wide spectrum of different fluorescent protein FRET pairs (*Table 1*; *Figure 1A*). Use of different fluorescent proteins can have large effects on the brightness and ligand-induced FRET changes of a biosensor (e.g. YFP vs improved variant Aphrodite (*Deuschle et al., 2006*). Wild-type fluorescent proteins can form dimers or multimers, and mutations along the dimerization interface of the proteins can reduce (e.g., monomeric GFP, mGFP, *Zacharias et al., 2002*) or promote dimerization (enhanced dimerization (ed) fluorescent protein variants). Enhanced dimerization variants have successfully been used for improved sensor design (e.g., *Vinkenborg et al., 2007*, *2009*). To allow for testing a large combinatorial space of sensory domains and FRET pairs, we engineered a destination vector series encoding 10 different fluorescent protein FRET pairs (*Table 1*). Fluorescent protein variants included brightness variants and dimerization variants, as well as N- or C- terminal truncation variants ((indicated as t7 or t9 for truncation of 7 or 9 amino acids), *Table 1*, *Supplementary file 1*). Initial experiments, in which constructs were expressed in *Escherichia coli* and yeast, were unsuccessful, most likely due to proteolysis of the fusion proteins, as evidenced by maintenance of fluorescence but loss of FRET during cell lysis (*Figure 2*). Proteolysis appeared specific to PAS domain sensors and was not observed for other FRET sensors (*Fehr et al., 2002*, *2003*; *Lager et al., 2003*; *Okumoto et al., 2005*;

**Table 1.** Guide to sensor component variants used for screening (for details see *Supplementary file 1*).

| pDR FLIP | FRET acceptor | N-term Linker | dPAS N-term | dPAS Linker | dPAS C-term | C-term Linker | FRET donor |
|---|---|---|---|---|---|---|---|
| | **Destination vector N-term** | **Gateway site** | **dPAS Entry clone** | | | **Gateway site** | **Destination vector C-term** |
| 30 | Aphrodite.t9 | | HAB1 · ABI1aid · ABI1cd · PYR1 · PYL1 · PYL4 · PYL5 · PYL6 · PYL7 · PYL8 · PYL9 · PYL10 | L12 Flexible · L52 Spring · L65 Flexible · L71 α-helix · L118 Spring & α-helix | HAB1 · ABI1aid · ABI1cd · PYR1 · PYL1 · PYL4 · PYL5 · PYL6 · PYL7 · PYL8 · PYL9 · PYL10 | | mCerulean |
| 32 | Aphrodite.t9 | | | | | | t7.eCFP.t9 |
| 34 | Aphrodite.t9 | | | | | | t7.TFP.t9 |
| 35 | Aphrodite.t9 | | | | | | mTFP.t9 |
| 36 | Aphrodite.t9 | attB1 invariant | | | | attB2 invariant | Cerulean |
| 37 | Citrine | | | | | | Cerulean |
| 38 | edCitrine | | | | | | edCerulean |
| 39 | edAphrodite.t9 | | | | | | t7.ed.eCFP.t9 |
| 42 | Citrine | | | | | | mCerulean |
| 43 | edAphrodite.t9 | | | | | | edCerulean |

Abbreviations: t7, t9: 7 or 9 amino acid terminal truncations; aid: ABA interaction domain; cd: catalytic domain; ed, m: enhanced dimerization or monomeric variant

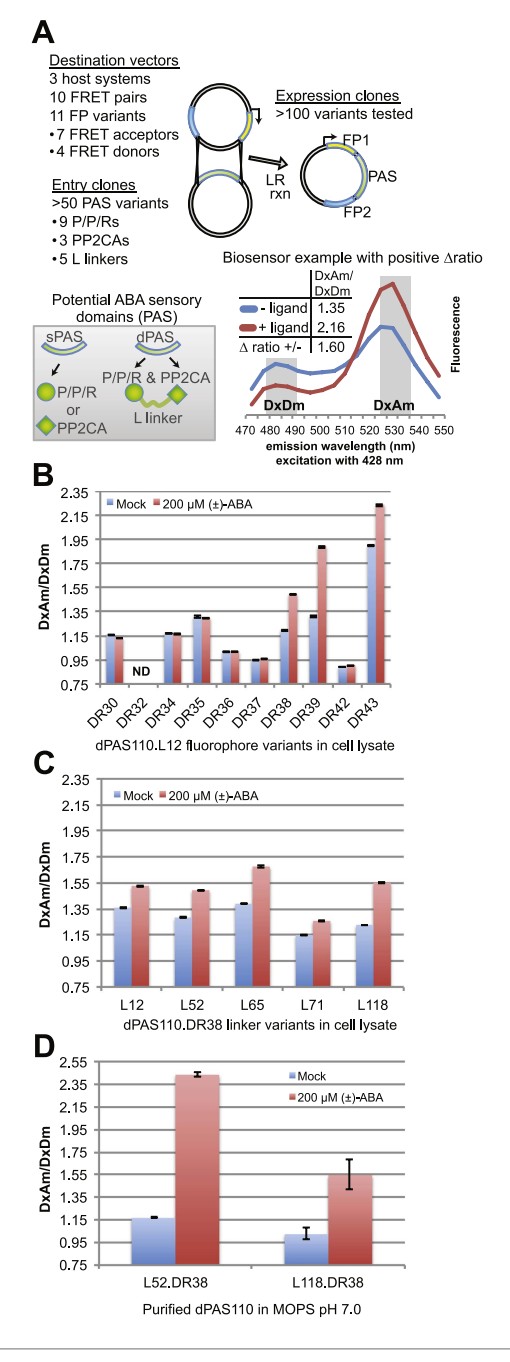

**Figure 1**. ABA responses of potential FRET sensors expressed in yeast and tested in yeast cell lysates or as purified proteins. (**A**) Diagram of cloning strategy based on pDR FLIP Destination vectors encoding FRET fluorescent protein pairs and PAS Entry clones encoding sPAS or dPAS ABA sensory domains. Also shown is an example of fluorescence emission curves without and with ligand for a sensor [ABACUS1-80μ, see below] with a positive ratio change (Δ DxDm/DxAm) of 1.6. (**B**) One linker variant (L12) of one dPAS (110) combined with nine fluorescent protein pairs. (**C**) Five linker variants of one dPAS combined with one

*Figure 1. Continued on next page*

*Gu et al., 2006*; *Lager et al., 2006*). Sensory domain proteolysis was circumvented by using a protease-deficient yeast strain (strain BJ5465 lacking Pep4 and Prb1, *Figure 2*) for expression and purification.

## Screening for ABA sensors

A first screen was conducted by analyzing ABA-dependent changes in fluorescence emission spectra of sPAS sensors in cell lysates cleared of cellular debris by centrifugation (*Figure 1B*, *Figure 1—figure supplement 1A*). FRET sensor responses were calculated as an ABA-induced change in the ratio of FRET acceptor emission (Am) to FRET donor emission (Dm) after excitation of the FRET donor (Dx); abbreviated ΔDxAm/DxDm. Four out of 15 sPAS sensors showed a relatively small ratio change (~1.06–1.12) in response to addition of 200 μM (±)-ABA (*Figure 1—figure supplement 1A*). We hypothesized that dPAS sensors containing both partners of an ABA co-receptor complex might exhibit larger ABA-dependent ratio changes due to larger conformational changes triggered by ABA-dependent intra-molecular interactions between two ABA binding domains. The presence of both ABA interaction domains may also increase ABA affinity, as demonstrated for PYR/PYL/RCAR affinity in the presence of PP2CA co-receptors (*Santiago et al., 2012*). An initial screen of dPAS containing full length PP2CA HAB1 linked via a flexible 12-amino acid linker (L12) rich in alanine, glycine and serine to the N- or C-terminus of nine different PYR/PYL/RCARs (dPAS19-dPAS36, *Supplementary file 1*) in cleared cell lysates identified several ABA-responsive candidates (e.g., dPAS20: N-terminal HAB1-L12-C-terminal PYL1; *Figure 1—figure supplement 1A*). We also screened dPAS designs with truncations of the PP2CA ABI1 in place of full length HAB1 (*Figure 1—figure supplement 1B*). Specifically ABI1 was truncated to its catalytic domain (ABI1cd) or a minimal ABA interaction domain (ABI1aid; 49 amino acids in length; *Supplementary file 1*). These variants (dPAS97-120, *Supplementary file 1*) constitute smaller proteins with potentially fewer interactions of the PP2CA domain with endogenous components in vivo; moreover, ABI1aid likely lacks PP2CA enzymatic activity. Among these combinations, ABI1aid fused to PYL1 (dPAS110) consistently yielded comparatively large ratio changes in response to ABA (ΔDxAm/DxDm ~1.29 to 1.63; *Figure 1—figure supplement 1B*). The screen was repeated with variants in which an artificial linker consisting of a molecular spring domain coupled to an α-helix was inserted between the two domains (L118; *Table 1*). Among them, the

*Figure 1. Continued*

fluorescent protein pair. (**D**) Two linker variants of one dPAS combined with one fluorescent protein pair tested as purified proteins. DxAm/DxDm = acceptor emission with donor excitation over donor emission with donor excitation. dPAS = double putative ABA sensory domain.

The following figure supplements are available for figure 1:

**Figure supplement 1**. Fluorescence emission curves and ABA response of potential FRET sensors expressed in yeast and tested in yeast cell lysates or as purified proteins (excitation wavelength = 428 nm).

ABI1aid/PYL1 combination again showed robust responses to ABA (*Figure 1—figure supplement 1B*). Although dPAS110 yielded biosensors with consistently high ratio change, similar sensory domains are also promising and could exhibit complementary properties. For example, PYR1 and PYL1 share 77% amino acid sequence identity and the use of dPAS109, containing PYR1 in an otherwise identical dPAS design, also led to ABA sensors with robust responses to ABA (*Figure 1—figure supplement 1B*). Furthermore, the use of ABI1cd/PYL1 (dPAS98) led to ABA responsive biosensors that could exhibit higher affinity for ABA compared to dPAS110 due to the presence of a complete PYL1 interaction interface in the ABI1 catalytic domain (*Figure 1—figure supplement 1B*). Many other dPAS variants showed low or no detectable ABA response in this initial screen (*Figure 1—figure supplement 1B*), underscoring the empirical nature of sensor development and the value of large-scale screens.

## ABA FRET sensor optimization

Linkers can have dramatic effects on sensor responses (*Deuschle et al., 2005b*; *Hires et al., 2008*; *Takanaga et al., 2008*). Therefore, five different linkers were inserted between the two PAS domains in dPAS110. In addition, we screened the five dPAS110 linker variants in fusions to nine different fluorescent protein pair variants (*Figure 1—figure supplement 1C*) in cleared cell lysates. Many of the sensor variants did not show responses to ABA addition, however FRET pair variants containing enhanced dimerization versions of the fluorescent proteins were consistently ABA responsive (*Figure 1B*, *Figure 1—figure supplement 1C*). As seen before, dPAS110 with L12 and L118 linkers showed large ABA-triggered responses when fused to the edAphrodite.t9/t7.ed.eCFP.t9 pair; notably the other three linker variants also showed ABA responses (*Figure 1—figure supplement 1C*). Similar effects of the linker variants were obtained for edCitrine/edCerulean (*Figure 1C*, *Figure 1—figure supplement 1C*). To obtain high quality data, sensor proteins were affinity purified in high throughput (*Figure 1—figure supplement 1D*) or individually (*Figure 1D*). dPAS110 fused by linkers L52 or L118 containing

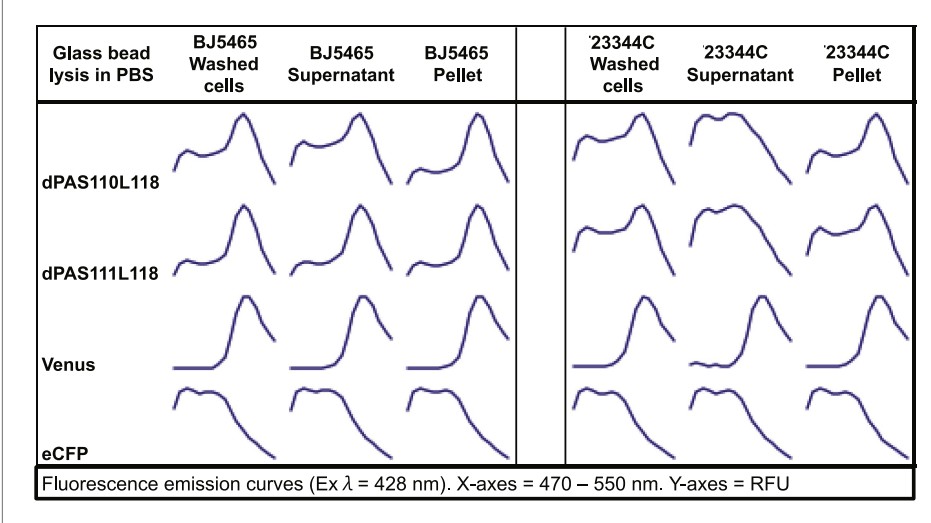

Fluorescence emission curves (Ex λ = 428 nm). X-axes = 470 – 550 nm. Y-axes = RFU

**Figure 2**. Fluorescence emission spectra of two sensors and individual fluorescent proteins expressed in yeast strain 23344C and protease deficient yeast strain BJ5465 (excitation wavelength = 428 nm). Proteins were analyzed in washed cells and in the supernatant or pellet of cell lysates.

the elastic GPGGA-repeat (i.e. molecular spring motif, *Grashoff et al., 2010*) and sandwiched by edCitrine/edCerulean yielded the highest ratio change (*Figure 1D*, *Figure 1—figure supplement 1D*). dPAS110.L52.DR38, a fusion protein harboring edCerulean/edCitrine (pDR FLIP38), the ABA interaction domain of a PP2CA ABA co-receptor (ABI1aid), an ABA receptor (PYL1), and three distinct linkers connecting the domains (attB1, L52, attB2), was named <u>Ab</u>scisic <u>A</u>cid <u>C</u>oncentration and <u>U</u>ptake <u>S</u>ensor version 1 (ABACUS1, *Figure 3A*). In ABACUS1, the binding of ABA to PYR/PYL/RCAR receptor proteins and the ABA-dependent interaction of PYR/PYL/RCARs with PP2CA phosphatases is co-opted to drive an ABA-dependent structural change in the fusion protein, which increases energy

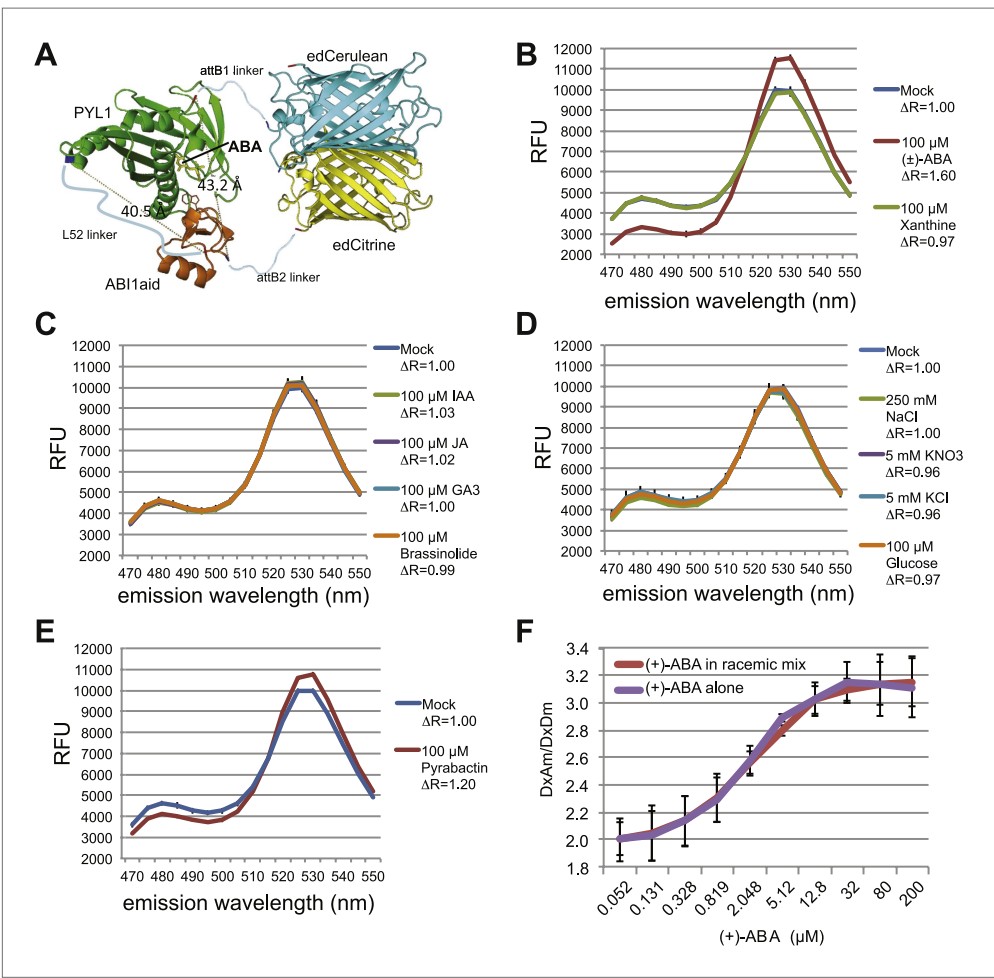

**Figure 3**. ABACUS1 design and fluorescence response to ABA and related compounds. (**A**) Hypothetical model of ABACUS1 bound to ABA. The structure shown for dPAS110 is derived from a crystal structure of ABA bound to PYL1 and ABI1 (PDB: 3JRQ) (*Miyazono et al., 2009*), and the structure shown for FRET donor enhanced dimer Cerulean (edCerulean) and FRET acceptor enhanced dimer Citrine (edCitrine) are derived from a crystal structure of *Aequorea victoria* GFP (PDB:1EMA) (*Ormö et al., 1996*). The structures are visualized using MacPyMol cartoon representation except for the ABA interacting tryptophan 300 of ABI1, which is shown in line representation and (+)–abscisic acid (yellow, ABA), which is shown in stick representation. The N-termini are colored blue and C-termini are colored red. The linkers are represented as hypothetical cartoon models not derived from known structures. The expected distance between the C-terminus of ABI1aid and the N-terminus of PYL1 (linked by L52 in ABACUS1) and the C-terminus of PYL1 and the N-terminus of ABI1aid (linked to fluorescent proteins), is shown in angstroms. The overall domain order of ABACUS1 is N-terminus–edCitrine–attB1–ABI1aid–L52–PYL1–attB2–edCerulean–C-terminus. ABACUS1-2μ fluorescence emission at shown wavelengths in response to (±)-ABA or xanthine (**B**), other phytohormones (**C**), glucose and various salts (**D**), pyrabactin (**E**). Excitation wavelength = 428 nm. Delta ratio (ΔR) = treatment DxAm/DxDm/mock DxAm/DxDm. (**F**) Titration curve for ABACUS1-2μ in response to equivalent concentrations of (+)-ABA supplied alone or as part of a racemic mixture with (−)-ABA.

transfer between donor and acceptor and results in a positive ratio change. This increase in energy transfer did not occur with the metabolic precursor xanthine, other plant hormones, or other ions and metabolites tested (*Figure 3B–D*), but was also induced by the ABA analog pyrabactin (*Figure 3E*). The ratio change was proportional to the ABA concentration (*Figure 4*); the prototype ABACUS1 is characterized by an apparent affinity of ~80 µM for (+)-ABA, and thus is referred to here as ABACUS1-80µ (*Figure 4*).

## Generation of ABA sensor affinity mutants

Since the cytosolic ABA concentration in individual cells is unknown, and because it is important to exclude artifacts caused by other parameters, an affinity series was generated. Structural studies of the ABA receptor protein complex guided the generation of ABACUS1-80µ affinity variants (*Melcher et al., 2009*; *Miyazono et al., 2009*; *Nishimura et al., 2009*; *Santiago et al., 2009*; *Hao et al., 2010*; *Melcher et al., 2010*; *Peterson et al., 2010*; *Mosquna et al., 2011*; *Santiago et al., 2012*; *Soon et al., 2012*; *Sun et al., 2012*; *Zhang et al., 2012*). Mutation of histidine-60 to proline switches PYR1 from a dimer to a monomer and increases its affinity for ABA (*Dupeux et al., 2011*); the corresponding mutation in the PYL1 moiety of ABACUS1-80µ produced ABACUS1-2µ, a variant with a higher affinity for ABA of ~2 µM (*Figure 4*). Disruption of the ABA binding pocket of PYL1 by mutation of the ABA coordinating lysine-86 to alanine abrogates ABA binding (*Melcher et al., 2009*); the corresponding mutation in ABACUS1-80µ resulted in a non-responsive variant ABACUS1-nr (*Figure 4*). Mutation of ABI1 tryptophan-300 to alanine abrogates its interaction with ABA and the PYL1 binding pocket (*Melcher et al., 2009*); the respective mutation in ABACUS1-80µ (ABACUS1-W300A) showed a reduced ratio change in response to ABA, but surprisingly no reduction in affinity (*Figure 4*). This finding indicates that ABA binding to PYL1 determines the apparent $K_d$, whereas the formation of the ABA-PYL1-ABI1 ternary complex is required for a maximal ratio change in ABACUS1-80µ. Finally, a histidine-142 to alanine mutation in the ABA binding pocket of PYL1, which is known to reduce ABA affinity (*Melcher et al., 2009*), resulted in a drastically reduced affinity in ABACUS1-H142A (*Figure 4*). The linear detection range of FRET sensors with a single binding site covers approximately two orders of magnitude; ABACUS1-2µ and ABACUS1-80µ together cover an overlapping range from ~200 nM to 800 µM (*Figure 4*). Further analyses focused on the use of these sensors. However, a more detailed analysis of the suite of sensors identified in the screen will allow for rapid optimization and expansion of the ABACUS toolbox, for example searches for combinations or variants with higher affinities and optimization against potential side effects of expression of the sensors on physiology.

## ABA uptake into yeast cells expressing ABA transporters

Yeast two-hybrid interaction screens can be used as an indirect way of monitoring ABA transport. When supplied exogenously with ≥10 µM ABA, the ABA uptake activity (through endogenous transporters) of yeast reaches high enough levels to allow detection of ABA-mediated yeast two-hybrid interactions between the ABA co-receptor components PYR/PYL/RCAR and PP2CAs

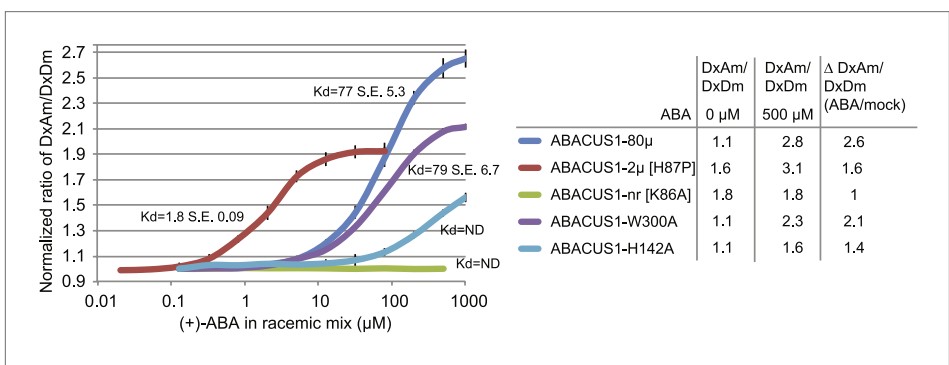

| | DxAm/DxDm | DxAm/DxDm | Δ DxAm/DxDm (ABA/mock) |
|---|---|---|---|
| ABA | 0 µM | 500 µM | |
| ABACUS1-80µ | 1.1 | 2.8 | 2.6 |
| ABACUS1-2µ [H87P] | 1.6 | 3.1 | 1.6 |
| ABACUS1-nr [K86A] | 1.8 | 1.8 | 1 |
| ABACUS1-W300A | 1.1 | 2.3 | 2.1 |
| ABACUS1-H142A | 1.1 | 1.6 | 1.4 |

**Figure 4**. ABA response titration curves for wild-type ABACUS1 (ABACUS1-80µ) and four different point mutants. DxAm/DxDm values for purified sensor samples plus ABA were normalized to corresponding mock treated samples. $K_d$ values were determined using Prism software (GraphPad) and are calculated based on the (+)-ABA concentration since ABACUS1 is stereospecific to (+)-ABA (*Figure 3F*).

(*Park et al., 2009*). At reduced ABA levels, the ABA-mediated interaction is not detectable (*Kanno et al., 2012*). A screen at low ABA levels had successfully been used to identify members of the nitrate/peptide transporter superfamily as putative ABA importers (*Kanno et al., 2012*). Here we directly tested the ability of an ABA transporter to mediate ABA accumulation in yeast cells using ABACUS1 sensors. Yeast cells did not accumulate sufficient ABA to induce a ratio change in ABACUS1-2μ even at 20 μM exogenous (±)-ABA (*Figure 5*). However, expression of the high affinity ABA importer AIT1/NRT1.2 conferred ABA transport activity as detected by DxAm/DxDm ratio change of ABACUS1-2μ and ABACUS1-80μ sensors (*Figure 5*). ABACUS1-2μ exhibited a near maximal ratio change in response to exogenous ABA supplied well below the sensor's apparent $K_d$ of 2 μM (+)-ABA, suggesting that AIT1/NRT1.2 concentrates ABA against an ABA gradient.

## ABACUS expression in Arabidopsis plants

FRET sensor expression under control of the CaMV 35S promoter has been shown to be subject to gene silencing (*Deuschle et al., 2006*; *Krebs et al., 2012*). Sensor silencing was reduced in the *rdr6-11* gene silencing mutant of *Arabidopsis* (*Peragine et al., 2004*; *Deuschle et al., 2006*). When FRET calcium sensors were expressed from the *UBQ10* promoter, silencing in wild type (Col0) plants was reduced (*Krebs et al., 2012*). Therefore, *ABACUS1* was expressed under control of the *UBQ10* promoter in both wild type and *rdr6-11* mutants of Arabidopsis (*Supplementary file 1*). However, as observed for other FRET sensors (*Deuschle et al., 2006*; *Chaudhuri et al., 2008*; *Lanquar et al., 2014*), significant levels of ABACUS1 fluorescence were detectable only in the *rdr6* silencing mutant. Confocal microscopy showed bright fluorescence from the sensors, which lacked subcellular targeting sequences, in the cytosol of roots, hypocotyls and leaves (root cell localization shown in *Figure 6A*). Seedlings of lines expressing *ABACUS1* affinity variants grown on ½ × MS medium agar plates or on soil were phenotypically indistinguishable from each other and from untransformed controls (*Figure 7B*, *Figure 7—figure supplement 1*). Primary root growth of ABACUS1 lines was not hypersensitive to inhibition by salt or osmotic stress (*Figure 7C*). In several experiments, ABACUS1-2μ lines showed increased anthocyanin accumulation in cotyledons under high salt conditions. However, primary root growth and germination of ABACUS1 lines was hypersensitive to inhibition by exogenous ABA, and hypersensitivity correlated with the affinity of the PYL1 domain of the biosensors (*Figure 7C–E*). For example, at 20 μM exogenous (±)-ABA, root growth of ABACUS1-2μ sensor lines was almost completely inhibited, while root growth of ABACUS1-80μ plants was strongly inhibited at 100 μM exogenous (±)-ABA. For comparison, the ABA root inhibition phenotype of a PP2CA triple mutant (*abi1-2*, *hab1-1*, *pp2ca-1*; *Rubio et al., 2009*), which is also hypersensitive to ABA, was similar to ABACUS1-80μ plants. The ABA hypersensitivity of ABACUS1 plants indicates that the PYL1 domain of ABACUS1 can inhibit endogenous PP2C co-receptor proteins as overexpression of PYR/PYL/RCAR proteins also results in ABA hypersensitivity phenotypes (*Mosquna et al., 2011*; *Pizzio et al., 2013*). These potential side effects can likely be overcome by making use of the suite of sensors developed in the sensor screen, specifically sensors that have a higher affinity and contain a less truncated PP2CA polypeptide (e.g., with dPAS20 or dPAS98 sensory domains), as well as variants that make use of other PYR/PYL/RCARs such as PYR1 in the dPAS109 sensory domain. Moreover, the existing sensors can be improved by mutagenesis of interaction domains for endogenous factors, as demonstrated by redesigning the binding interface of calmodulin and the calmodulin-binding peptide to produce calcium sensors that do not interact with endogenous calmodulin (*Palmer et al., 2004*). For example, mutation of S112 in the ABACUS PYL1 domain

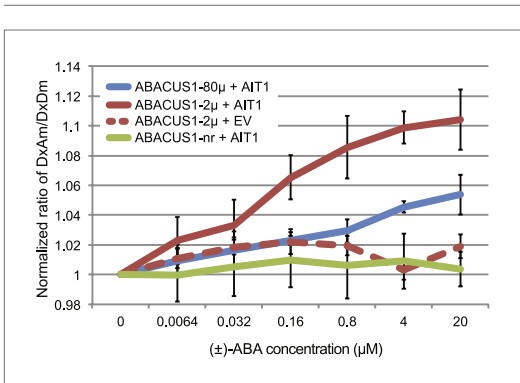

**Figure 5**. ABA import by AIT1/NRT1.2 in protease deficient yeast expressing ABACUS1 variants. ABA was added to yeast cells suspended in 20 mM MES buffer (pH4.7) and ABACUS1 fluorescence was measured after 5 min. DxAm/DxDm values for yeast samples plus ABA were normalized to corresponding mock treated samples. EV = empty vector control. Because ABACUS1 is stereospecific for (+)-ABA the effective ABA concentration available for sensing is ½ the (±)-ABA concentration.

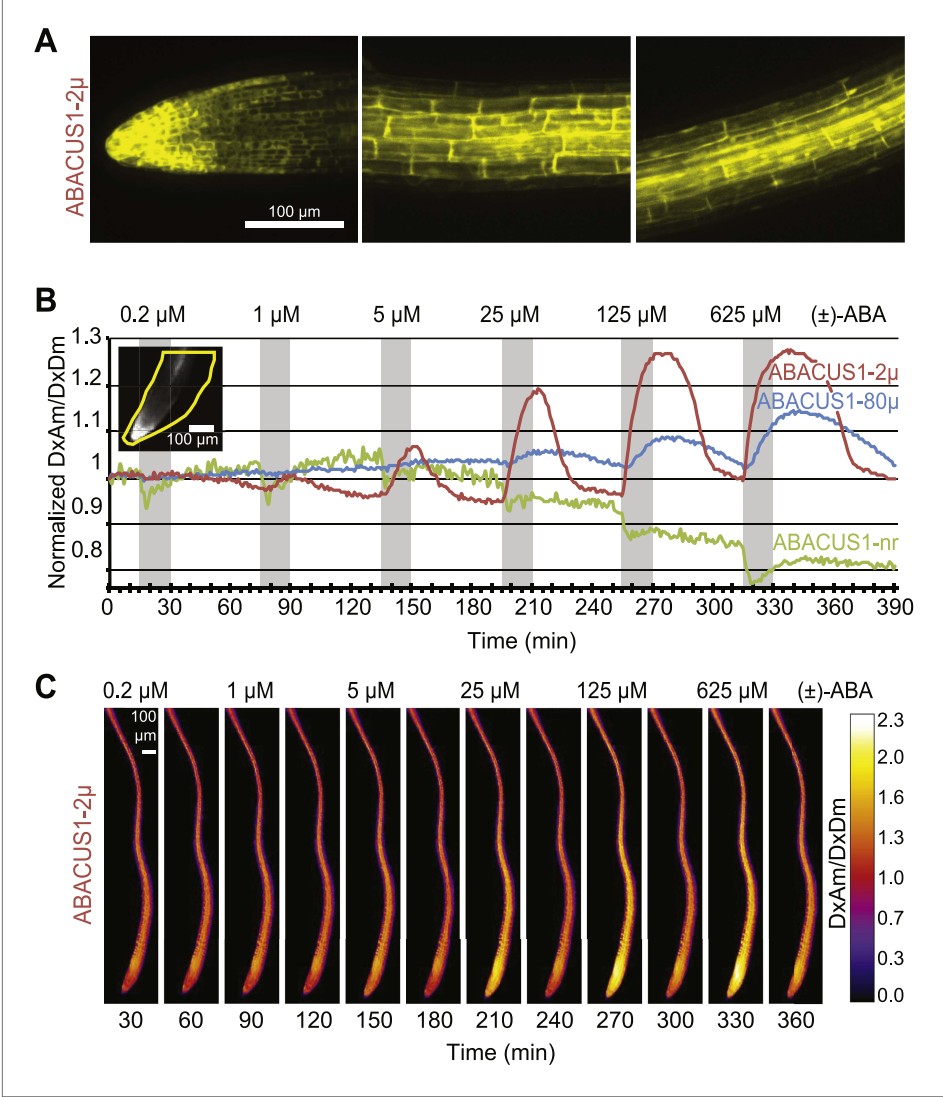

**Figure 6**. Expression pattern and ABA responses of ABACUS1 in Arabidopsis roots. (**A**) Expression pattern and cytosolic localization of ABACUS1-2μ fluorescence (AxAm) in root cells in the root tip (left), in the elongation zone (center), and in the differentiation zone (right). (**B**) ABA titrations of ABACUS1 in roots. Traces show ratio (DxAm/DxDm) changes of ABACUS1-2μ, ABACUS1-80μ and ABACUS1-nr roots in response to six 15-min pulses with increasing concentration of (±)-ABA. (±)-ABA pulses were raised in 5 × increments, from 0.2 μM to 625 μM. Inset shows the ABACUS1-2μ root at time zero with the region used for measurements outlined in yellow. ABA pulses are shown as grey areas and all ratios were normalized to time point 0. (**C**) Spatial distribution of ABA mediated responses. Ratio images showing pattern of ABACUS1-2μ in response to six (±)-ABA pulses as described above. Right: look up table used for false coloring of ratio images. Because ABACUS1 is stereospecific for (+)-ABA the effective ABA concentration available for sensing is ½ the (±)-ABA concentration.

The following figure supplements are available for figure 6:

**Figure supplement 1**. ABA titrations of ABACUS1 roots from *Figure 6B*.

would likely reduce interaction with endogenous PP2CAs without affecting interaction with the ABI1aid.

## Analysis of ABA dynamics in plant roots

Three ABACUS1 sensors were used to measure cytosolic ABA accumulation in response to addition of exogenous ABA to roots growing in the RootChip (*Grossmann et al., 2011*, *2012*) (*Figure 7A*). The

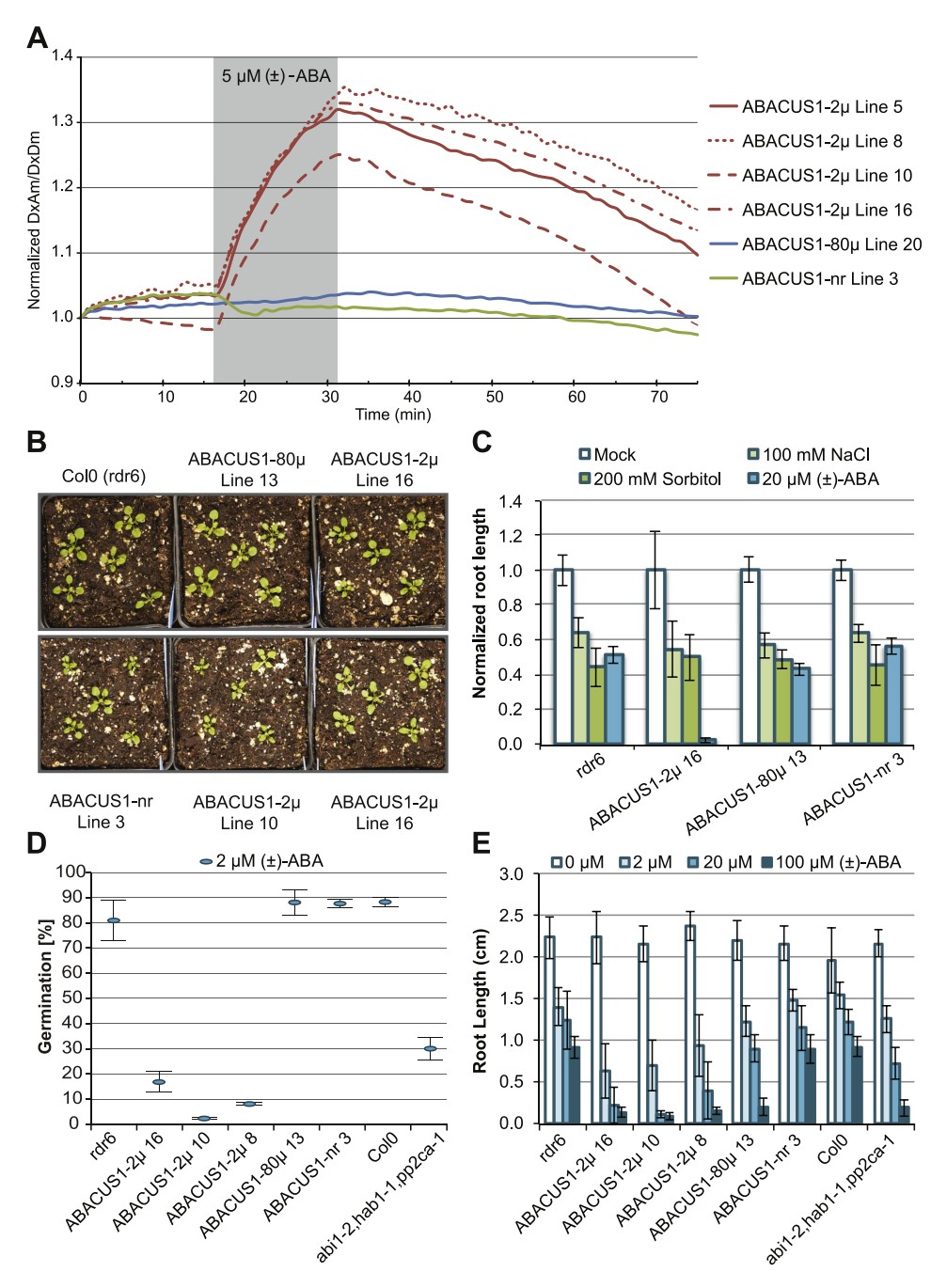

**Figure 7**. ABA responses and plant growth of ABACUS1 plant lines (homozygous, T3 generation). (**A**) Fluorescence response of ABACUS1 in roots to 5 μM (±)-ABA. All lines were imaged simultaneously while growing in the same RootChip16. Because ABACUS1 is stereospecific for (+)-ABA, the effective ABA concentration available for sensing is ½ the (±)-ABA concentration. (**B**) 3 week old ABACUS1 plant lines (homozygous T3) grown on soil in a long day light chamber. (**C**) Primary root growth of vertically grown Arabidopsis seedlings 4 days after transfer to ½ × MS agar medium plates and plates supplemented with 100 mM NaCl, 200 mM sorbitol or 20 μM (±)-ABA. Values are normalized to mock treated roots and are presented as means and standard deviations of ~15 roots. (**D**) Seed germination rate of Arabidopsis seeds sown on ½ × MS agar medium plates supplemented with 2 μM (±)-ABA. Germination percentage is presented as mean and standard error of the mean of three independent experiments of 100–200 seeds each. (**E**) Primary root growth of vertically grown Arabidopsis seedlings 4 days after transfer

*Figure 7. Continued on next page*

*Figure 7. Continued*

to ½ × MS agar medium plates with increasing concentrations of (±)-ABA. Root lengths are presented as means and standard deviations of ~15 roots.

The following figure supplements are available for figure 7:

**Figure supplement 1:**. Expression and Plant growth of ABACUS1 plant lines (homozygous T3 generation).

RootChip is a microfluidic device that permits environmental manipulations during continuous imaging of growing *Arabidopsis* roots. To analyze multiple treatments in different ABACUS1 lines simultaneously—required for side-by-side comparison of ABACUS1 behavior in different conditions–we used the second generation RootChip16, in which the chip architecture was modified to accommodate 16 individual roots in 14 mm long observation chambers, and which allows perfusion with up to seven different solutions per chamber or parallel perfusion of 2 × 8 roots with two different solutions (*Figure 8*).

Time-courses under perfusion with pulses of increasing ABA concentration revealed that ABACUS1-80μ and ABACUS1-2μ responded with positive ratio changes—up to ~1.3 for ABACUS1-2μ– to exogenous ABA in a reversible and concentration-dependent manner (*Figure 6B,C*), indicating that basal endogenous ABA levels did not saturate the sensors. No dose dependent positive ratio change was observed in plants expressing the control sensor ABACUS1-nr (*Figure 6B*). We observed that sensor brightness increased by about twofold within 6 hr after exposure of roots to ABA (*Figure 6— figure supplement 1*). Because sensors were expressed from the constitutive *UBQ10* promoter, the most parsimonious interpretation is that sensor transcripts or polypeptides are stabilized by ABA. It is conceivable that PYL1 and/or the ABA interaction domain of ABI1 (ABI1aid) are subject to ABA-controlled turnover. Identification of the underlying signaling mechanism could guide the engineering of ABACUS sensors with increased stability of expression and potentially be exploited to engineer ABA signaling sensors whose intensity is ABA responsive. Additionally, the use of alternate PYR/PYL/RCAR isoforms (e.g., potentially PYR1 in dPAS109) or targeting to compartments that are inaccessible to the degradative mechanism might eliminate interference by degradation.

Given that AIT1 as well as ABCG40, an ABC transporter identified as a putative ABA importer, function as primary or secondary active systems, one may expect ABA to accumulate in the root cell

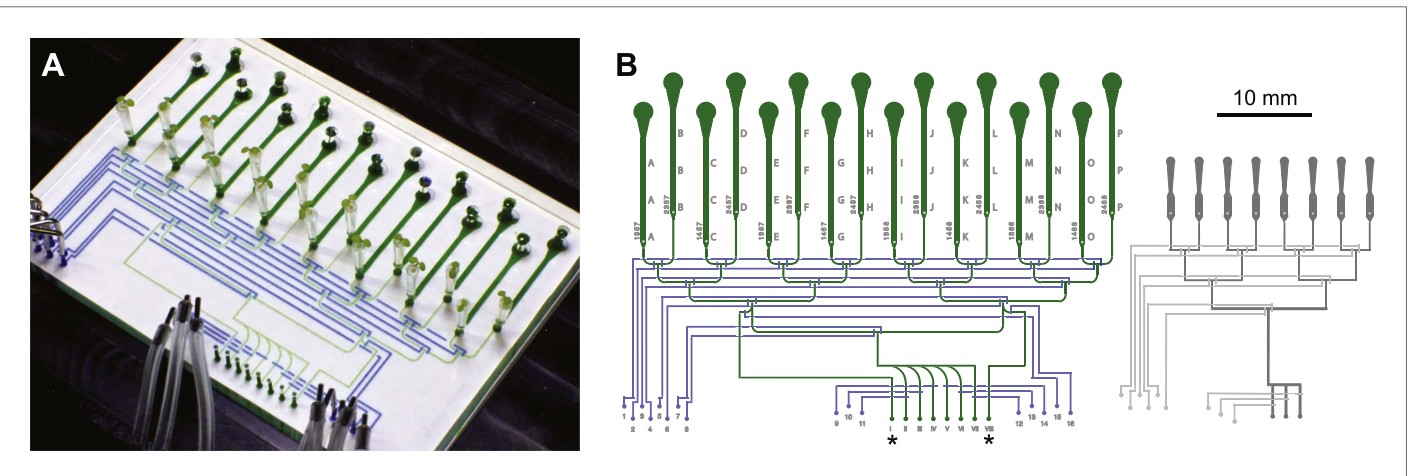

**Figure 8**. Architecture of the RootChip16. (**A**) Photograph of the RootChip16 with control layer microchannels and micromechanical valves stained in blue, flow layer with observation chambers in green, and 5-days old seedlings growing on medium-filled, cut pipette tips. (**B**) The device design accommodates 16 individual roots in 14 mm long observation chambers and features 6 solution inlets that address all chambers plus 2 inlets (asterisks) that allow perfusion of each half of the chip with two different solutions. To aid quick identification of chip features during imaging, labels are embossed as part of the control layer and visible under the microscope. Chambers are labeled with letters A–P and inlets are labeled with roman (solution inlets) or arabic numbers (valve microchannels). On the left of each chamber the combination of valves is visible that is required to direct the flow to the respective chamber. Right: for comparison, the RootChip design is drawn to scale.

cytosol against a concentration gradient. However, assuming that the affinity of the sensor is similar under in vitro and in vivo conditions, ABA does not appear to accumulate in the cytosol above externally supplied levels. Rather, at low micromolar concentrations, cytosolic levels approximate external levels (the response $K_{0.5}$ is ~4 µM (+)-ABA with ABACUS1-2µ, *Figure 6B*). It is, however, important to note that steady state ABA levels in the cytosol also depend on the rates of compartmentation and degradation, and therefore the ability of a transport system to accumulate ABA is necessarily underestimated. Transport might also be underestimated by about twofold if transport activity is limiting and non-stereoselective for (+)-ABA (i.e., competition from (−)-ABA limits apparent (+)-ABA import) since roots were treated with (+)-ABA as part of a racemic mix, and ABACUS1 is stereoselective for (+)-ABA (*Figure 3F*). It is noteworthy that ABA levels accumulated much more slowly compared to glucose (*Chaudhuri et al., 2008*); saturation of sensor responses was reached only after ~15 min in a typical experiment compared to less than 1 min for glucose (*Grossmann et al., 2011*) indicating that the uptake capacity for ABA is significantly lower relative to glucose. In contrast to ABACUS1-2µ, which saturated between 12.5 and 62.5 µM external (+)-ABA, ABACUS1-80µ continued to respond to higher levels of ABA (i.e., 312.5 µM (+)-ABA, *Figure 6B*). If we assume that in vivo the two sensors retain their relative in vitro ΔDxAm/DxDm properties (*Figures 4*), ABACUS1-80µ reaches ~30% of ABACUS1-2µ maximal ratio change at 312.5 µM external (+)-ABA, which would correspond to an (+)-ABA concentration between 12.5 and 62.5 µM (i.e., <<312.5 µM). Thus, at higher concentrations of (+)-ABA (i.e., >62.5 µM), cytosolic levels do not approximate external levels and the import of ABA is partly saturated. Overall, the responses of cytosolic ABACUS1-2µ and ABACUS1-80µ indicate that micromolar levels of externally supplied ABA are imported slowly and not concentrated in root cells, characteristics that are not necessarily consistent with known mechanisms of ABA import (i.e., ion-trap, AIT1, ABCG40). Furthermore, the pattern of ABA import in different root regions and at various developmental stages appeared similar (*Figure 6C*, *Figure 9A*; *Video 1*), inconsistent with the observed expression patterns of known ABA importers (*Figure 10*). The ABA transport characteristics detected with ABACUS sensors are best explained by the activity of a broadly expressed ABA uniporter that has still to be identified.

## ABA pretreatment affects kinetics of ABA accumulation

ABA synthesis, degradation and transport are highly regulated, for example the gene for the key enzyme for ABA catabolism CYP707A is induced by ABA (*Kushiro et al., 2004*). The kinetics of FRET sensor responses can provide insights into the relevant metabolic fluxes. When pulsing roots with identical amounts of ABA, the slopes of ABACUS1-2µ responses changed for both accumulation and elimination phases (*Figure 9A*). The apparent rate change was further tested directly by comparing two sets of roots growing in the same RootChip, one set that received an initial pulse after 15 min, followed by two additional pulses at 75 and 135 min, and a second set that was not exposed to the first pulse (*Figure 9B*). Plots of the first derivatives show that during the first ABA pulse, the accumulation rate of a non-pre-exposed root was nearly twice that of a root pre-exposed to ABA. This difference may be caused by ABA triggering its own elimination from the cytosol, as the increase in elimination rate that occured gradually in the minutes following the first ABA pulse largely accounted for the decrease in accumulation rate at the start of the following ABA pulse. Interestingly, different zones of the root showed different elimination rates after a first ABA pulse, but all zones examined converged upon an accelerated elimination rate by a third ABA pulse (*Figure 9A*). This difference in initial elimination rate could largely explain the initial differences in ABACUS1-2µ responses between different zones, for example after receiving an ABA pretreatment, the root tip and elongation zone showed similar maximal ABA levels (*Figure 9A*). Since ABA degradation rates are expected to contribute to both decreased accumulation and increased elimination velocities, our data intimate the existence of an ABA-triggered induction of ABA degradation, modification or compartmentation, thus leading to reduced cytosolic ABA levels.

## ABA accumulation in leaves

ABA accumulation in leaf tissues regulates foliar responses to water deficit and osmotic stresses. Cell-specific transcriptomic analyses revealed that the expression of ABA biosynthetic enzyme genes after drought stress of Arabidopsis leaves was induced first in the vascular parenchyma followed by mesophyll cells (*Endo et al., 2008*). However, transcriptional and genetic evidence supports a role for guard cell autonomous ABA in reducing stomatal aperture under low humidity (*Bauer et al., 2013*). Dissection

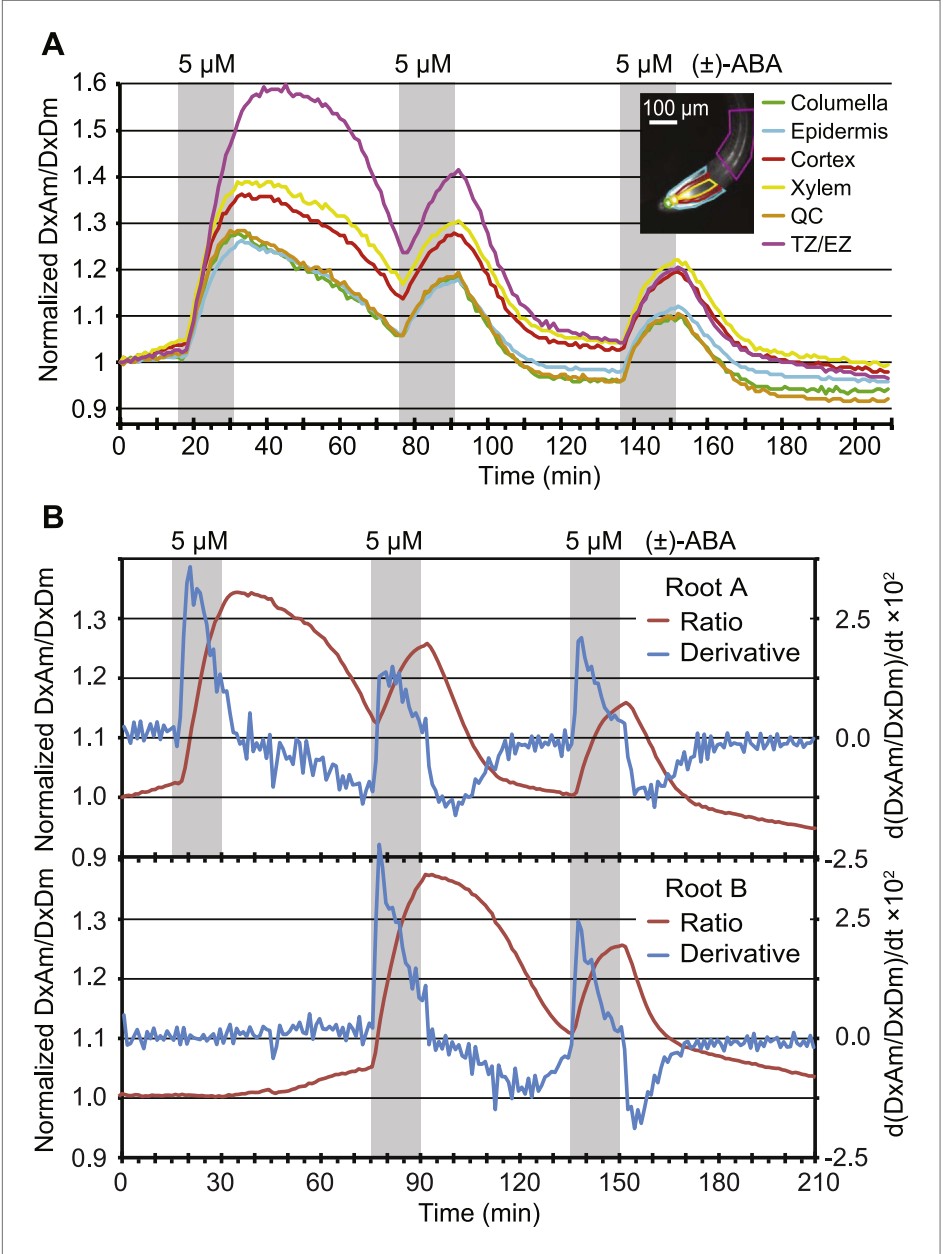

**Figure 9**. Dynamics of ABA uptake and elimination in roots as measured by ABACUS1-2μ. (**A**) Spatial differences in the dynamics of ABA response. Traces showing ratio (DxAm/DxDm) changes in different root zones of an ABACUS1-2μ expressing root in response to three 15-min pulses of 5 μM (±)-ABA. The insert shows the root at time 0 with the regions used for the different zones outlined in the respective colors. (**B**) Temporal differences in ABA response dynamics. Red traces show ratio (DxAm/DxDm) changes of two ABACUS1-2μ roots from an experiment where roots were exposed to either three (Root A, upper) or two (Root B, lower) consecutive pulses of 5 μM (±)-ABA. The blue lines show the derivative of the respective traces (d[DxAm/DxDm]/dt) to aid in interpreting the dynamics of uptake and elimination. For all traces, (±)-ABA pulses are shown as grey areas and all ratios were normalized to time point 0. Because ABACUS1 is stereospecific for (+)-ABA, the effective ABA concentration available for sensing is ½ the (±)-ABA concentration.

of water-stressed *Vicia faba* leaves and quantitative ABA analyses indicated that ABA accumulates in epidermal, mesophyll and guard cells to ~10 μM when ignoring compartmentation (*Harris et al., 1988*). To investigate ABA accumulation in the cytosol of leaf cells, we measured DxAm/DxDm ratios in ABACUS1-2μ Arabidopsis leaves 24 hr after transfer of the petioles of detached leaves to medium

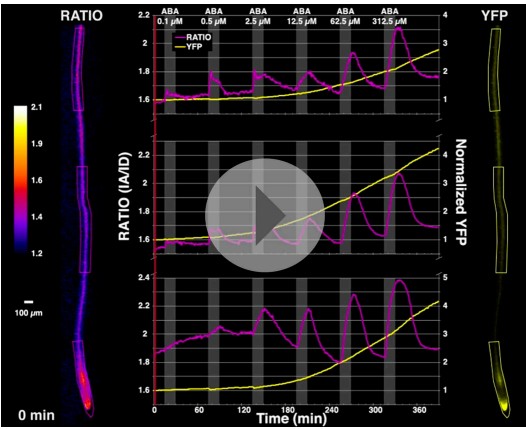

**Video 1**. The video shows an ABACUS1-2μ Arabidopsis root, growing in a RootChip16, exposed to pulses of increasing concentrations of ABA. The RATIO and YFP videos show the root throughout the experiment whereas the central charts show the average ratio (in magenta) and intensity (in yellow) of three regions (ROIs shown as overlaid on the root). The RATIO video (left) shows the ratio (DxAm/DxDm) using the color-code shown in the calibration bar on the leftmost side. The YFP video (right) shows acceptor emission intensities upon acceptor excitation (AxAm). Grey areas in the charts indicate ABA pulses of increasing concentrations (0.1–312.5 μM (+)-ABA, in 5x increments and supplied as racemic mix). The red vertical line shows position of current time in the charts. Scale bar corresponds to 100 μm and the counter in the lower left corner shows time elapsed in minutes.

supplemented with 150 mM NaCl. When compared to leaves transferred to a control solution, salt treated ABACUS1-2μ leaves showed increased average ratio (Δ ratio across five experiments between ~1.14 to 1.59, *Figure 11A*). In comparison, transfer to ABA medium (100 μM (±)-ABA) resulted in a greater increase in average ratio compared to controls (Δ ratio across five experiments between ~1.30 to 1.95, *Figure 11A*). To control for potential artifacts resulting from changes to the cellular environment upon salt stress on ABACUS1 (e.g., by increased solute concentrations), we also measured DxAm/DxDm ratios in ABACUS1-80μ and ABACUS1-nr leaves 24 hr after transfer to osmotic stress, ABA and control solutions (*Figure 11B,C*). Salt stress treatments of ABACUS 1–80μ leaves also increased average ratio (Δ ratio between 1.11 and 1.41) while salt effects on ABACUS1-nr average ratio were variable (*Figure 11B,C*). In contrast, the pattern of ABACUS1 responses to ABA treatments roughly corresponded to the Kd of the respective sensors. Overall, the variability of ABACUS1 leaf ratios in response to salt stress precludes interpretation of the NaCl induced Δ ratio in ABACUS1-2μ leaves as resulting from accumulation of ABA. Use of improved experimental approaches, ideally confocal fluorescence microscopy or light sheet microscopy of intact aerial tissues, as well as further optimized ABACUS1 lines, may help in elucidating spatiotemporal patterns of ABA levels in the leaf.

## Nuclear localized ABACUS is ABA responsive

To be able to improve analysis of spatial differences in roots as well as potential differences in ABA levels between cytosol and nuclei, ABACUS1-2μ was fused to a nuclear targeting sequence. Analysis of plant lines expressing nuclear localized ABACUS1-2μ[NLS] showed preferential nuclear localization as compared to ABACUS1-2μ (*Figure 12A–F*). Initial experiments in the RootChip16 demonstrated that the predominantly nuclear-localized sensor also responded to external ABA (*Figure 12G*). The nuclear-targeted sensor lines are expected to facilitate discrimination of ABA levels in neighboring cells.

## Discussion

The phytohormone ABA serves as a master regulator of plant stress avoidance and tolerance. Due to the extent and complexity of ABA signaling and ABA-dependent physiology, a more detailed understanding of the site and timing of ABA accumulation is critical. Here we show that FRET sensors for ABA, termed ABACUS, permit the investigation of ABA dynamics with high spatial and temporal resolution. Time course analysis of ABA perfusion of growing Arabidopsis roots expressing *ABACUS1* reveals broad, concentration-dependent, and reversible cytosolic and nuclear accumulation of ABA, and indicates that ABA treatment accelerates ABA metabolism.

The engineering of the ABACUS FRET sensors was a partially empirical process that required the iterative expression and testing of over 100 fusion proteins consisting of several variable component domains (i.e., a FRET pair of fluorescent proteins, ABA binding domain(s), and variable linker regions). The first highly ABA responsive ABACUS consists of edeCFP as the FRET donor, edAFP as the FRET acceptor, and a composite ABA sensory domain (dPAS110) consisting of a truncated ABI1 co-receptor (limited to ABA interaction domain) linked to the receptor PYL1, *Figure 3*). This specific composite ABA sensory domain (ABI1aid, flexible 12 amino acid linker, PYL1) resulted in ABA sensors with higher

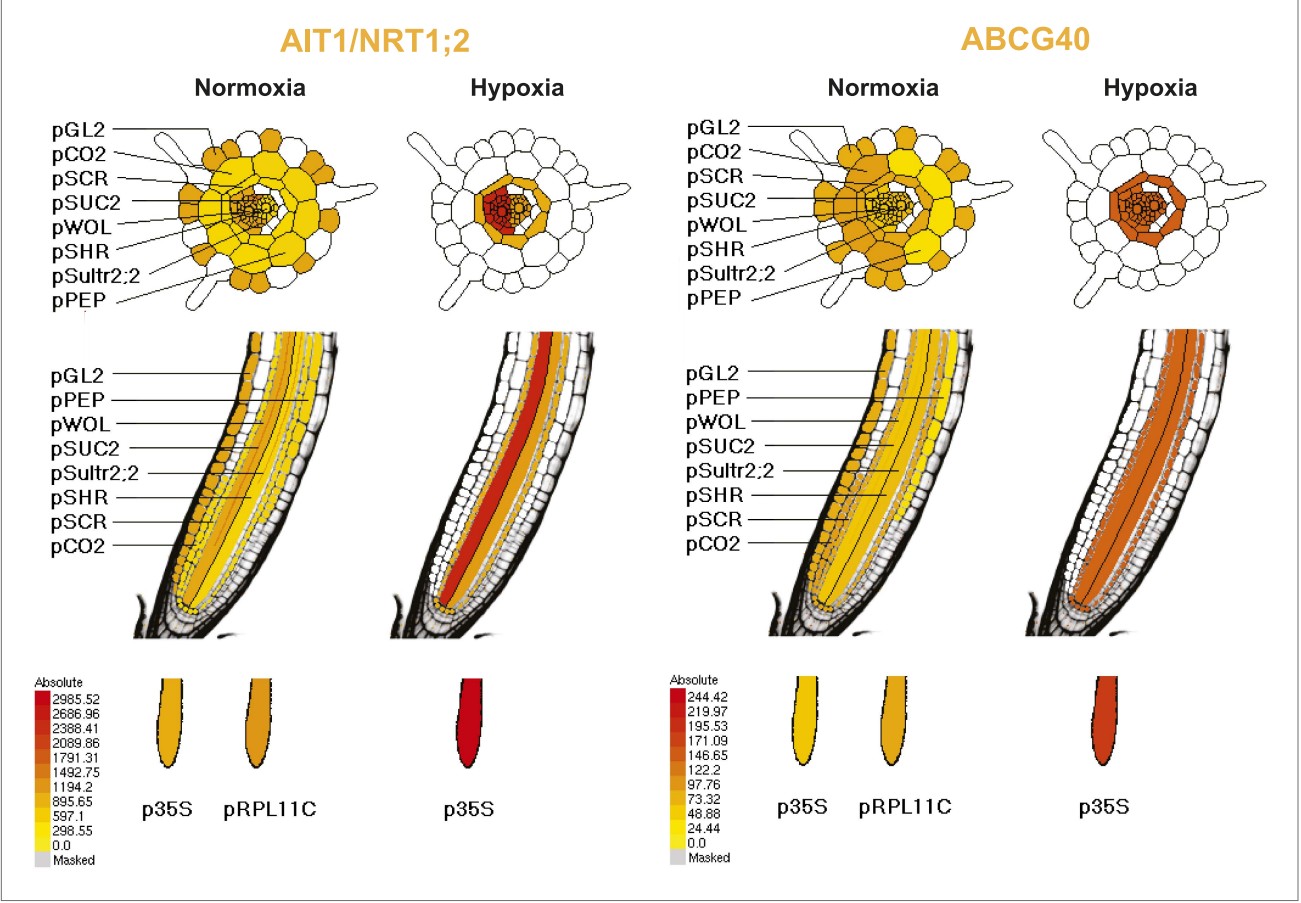

**Figure 10**. Pattern of ribosome-associated transcripts for AIT1 and ABCG40 transporter genes. Data are derived from microarray studies of RNA bound to polysomes (http://efp.ucr.edu/) (***Mustroph et al., 2009***). Cell-type-specific expression is based on coexpression with any of the six genes whose promoters were used for driving the ribosomal affinity tag: pGL.2 for trichomes, pCER5 for epidermis, pRBCS for mesophyll, pSULTR2.2 for bundle sheath, pSUC2 for companion cells and pKAT1 for guard cells. While the cell-specificity of the pSUC2 promoter is unambiguous in companion cells with leakage into the sieve elements (***Truernit and Sauer, 1995***), bundle sheath expression of pSULT2.2 is not as well documented (***Takahashi et al., 2000***). Further analysis is required, therefore the representation in this figures as taken from http://efp.ucr.edu/ may not accurately reflect the cell type specificity. The expression levels are color-coded with *yellow* indicating low levels of expression and *red* corresponding to high levels of expression.

ratio change compared to sensors with different sensory domain components (e.g., other PYR/PYL/RCARs or other truncations of ABI1), or different sensory domain configuration (e.g., PYL1, flexible 12 amino acid linker, ABI1aid). Truncation of ABI1 to a hypothetical minimal ABA interaction domain was selected as the 49 amino acid domain (H279–D327) that is flanked by, but is structurally distinct from, two parallel arrays of β-sheets that form the PP2CA catalytic domain (***Miyazono et al., 2009***). In addition to yielding large ratio changes relative to other dPAS component PP2CA domains, this truncation was deemed advantageous for its small size and for its presumed lack of PP2CA activity.

Initial ABACUS designs were further optimized through screening additional FRET pairs and linkers. The screen identified fluorescent proteins with enhanced dimerization tendency as the most likely to produce highly responsive ABA sensors (i.e., large ABA-dependent ratio changes). This modification had proven effective for increasing the FRET ratio change, and thus signal-to-noise ratio, in other sensors (***Vinkenborg et al., 2007***, ***2009***). The L52 linker consisting of an elastic GPGGA repeat (***Grashoff et al., 2010***) showed improved response to ABA in combination with multiple fluorescent protein pair combinations (***Figure 1***, ***Figure 1—figure supplement 1***). The enhanced ratio change observed for the increased dimerization fluorescent proteins and spring linker could both be attributed to an increase in the change of the relative fluorophore positioning between unbound and

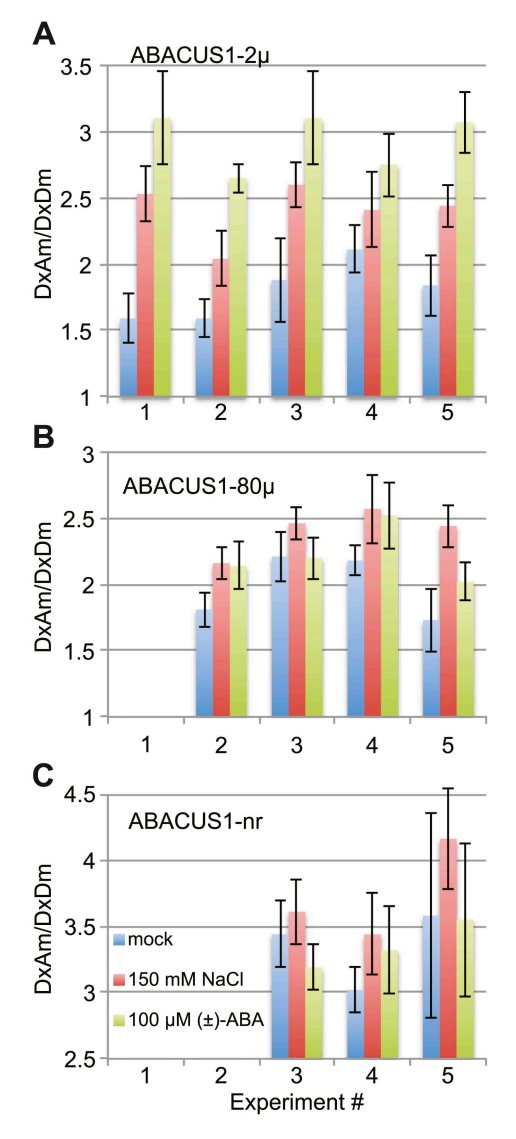

**Figure 11**. ABACUS1 response to salt and ABA treatment of detached leaves. DxAm/DxDm ratios for Arabidopsis leaves after 24 hr of suspension of the petiole in ¼ × MS medium or the same medium supplemented with 150 mM NaCl or 100 µM (±)-ABA. Average ratio are shown for (**A**) ABACUS1-2µ (**B**) ABACUS1-80µ and (**C**) ABACUS1-nr leaves. Because ABACUS1 is stereospecific for (+)-ABA the effective ABA concentration available for sensing is ½ the (±)-ABA concentration.

ABA-bound states according to the following model. The tension provided by the spring linker could enhance an 'open' state in which ABI1aid and PYL1 domains are apart, thereby reducing the FRET efficiency of the fluorophore pair. Upon ABA binding, ABI1aid and PYL1 domains interact and effect a 'closed' state, bringing the fluorophores into closer proximity and increasing FRET efficiency. Upon closure, fluorescent proteins with enhanced dimerization tendencies will enhance the closed state of the sensor through dimerization. Furthermore, the parallel orientation of the heterodimer state is expected to result in high FRET efficiency (*Figure 3A*). Verification of this model in which linker tension promotes an open state with lower FRET efficiency and fluorescent protein dimerization promotes a closed state with high FRET efficiency will require structural analyses. The ABACUS optimization process was accelerated using a cloning, expression and purification platform that is readily applicable to the engineering of FRET sensors for other small molecules or molecular events (*Table 1*; *Figure 1*). For example, the platform accelerated the development of the NiTrac nitrogen transensors (*Ho and Frommer, 2014*).

A series of ABACUS sensors with differing affinities for ABA was generated through site directed mutagenesis of residues critical for ABA binding (*Figure 4*). The apparent $K_d$ of 80 µM for the ABACUS1-80µ sensor and 2 µM for the ABACUS1-2µ sensor corresponds well with the $K_d$ measured in vitro of the PYL1 components of their respective ABA sensory domains (i.e., PYL1 and PYL1H87P, *Dupeux et al., 2011*). However, ABA co-receptor complexes between a PP2CA and a PYR/PYL/RCAR can have >10-fold higher affinity according to a model derived from in vitro measurements (*Dupeux et al., 2011*). Because the W300A mutation of the ABI1aid in ABACUS1-80µ did not affect the affinity (*Figure 4*), it is likely that, in contrast to ABI1, the ABI1aid does not increase PYL1 affinity for ABA. Thus, biosensors with less truncated PP2CA domains, for example in the dPAS20 and dPAS98 sensory domains (*Figure 1*, *Figure 1— figure supplement 1A,B*), could have higher affinities for ABA and be highly useful for measurements of lower ABA concentrations. Furthermore, the ABA sensors developed in parallel by *Waadt et al. (2014)* are also be complementary to the ones described here.

*ABACUS1-80µ*, *ABACUS1-2µ* and *ABACUS1-nr* were stably expressed in *Arabidopsis* plants and these plants were used to investigate the dynamics of cytosolic and nuclear ABA levels. It is noteworthy that these first generation ABA sensors have two potential drawbacks, that is their overexpression leads to ABA hypersensitivity and the sensor amount in roots was affected by ABA. Next generation ABACUS variants carrying mutations with reduced potential to interact with endogenous ABA

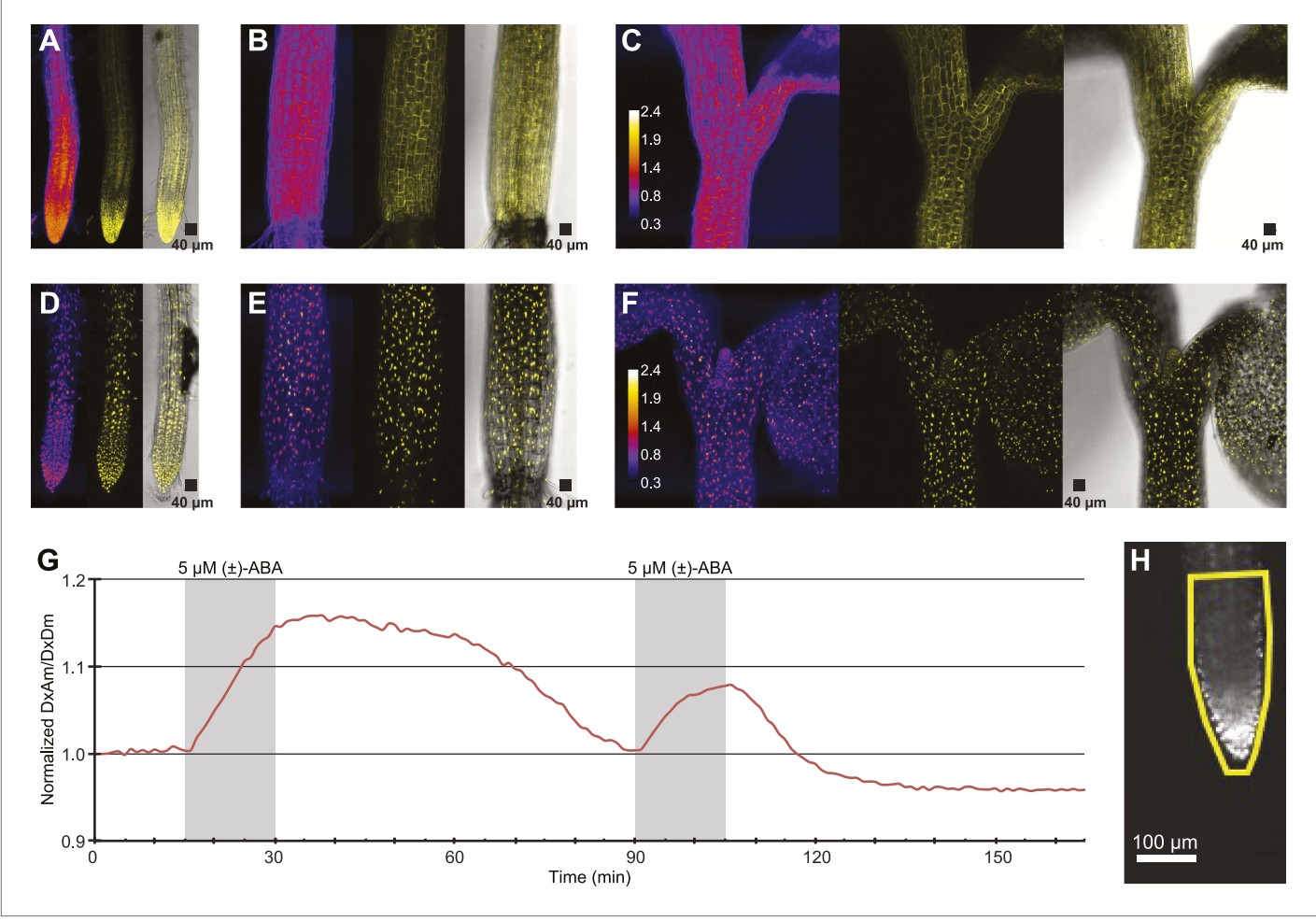

**Figure 12**. ABA response of nuclear localized ABACUS. (**A–F**) Confocal scanning microscopy images of the roots (**A** and **D**), hypocotyl (**B** and **E**) and region around the shoot apical meristem (**C** and **F**) of seedlings expressing either the cytosolic (**A–C**) or the nuclear localized (**D–E**) version of the ABACUS1-2µ sensor. From left to right the images in each panel show (Left) ratios (DxDm/DxAm, calculated as the sum of DxDm divided by the sum of DxAm in order to minimize artifacts due to pixel to pixel noise in z-direction), (Middle) maximum projections of the YFP signal and (Right) a bright field image of the middle plane in the z-stack with the maximum projected YFP signal overlaid. The calibration bar in ratio images (**C** and **F**) shows the look up table used for all ratio images. The scale bar in the bright field image with overlaid YFP shows a 40 µm × 40 µm square. (**G**) Trace showing ratio (DxAm/DxDm) of the root tip of a seedling expressing the nuclear localized ABACUS1-2µ sensor in response to ABA. The grey areas indicate two 15-min pulses of 5 µM (±)-ABA and the ratio is normalized to the time point 0. (**H**) The root imaged for the trace showed in (**G**) with the ROI used for measuring outlined in yellow and a scale bar showing the distance of 100 µm. Because ABACUS1 is stereospecific for (+)-ABA, the effective ABA concentration available for sensing is half the (±)-ABA concentration.

signaling components and variants that are unaffected in sensor synthesis/turnover will be required for more detailed physiological analyses. In experiments carried out in the RootChip, the transport of exogenous ABA into the cytosol of *Arabidopsis* root cells, as determined using ABACUS responses, occurred in all root zones measured (**Figure 6C**, **Figure 9A**; **Video 1**). We did not observe large differences in ABA import across tissues, as described by Astle and Rubery (**Astle and Rubery, 1980**, **1983**), potentially because sensor measurements can accurately reflect cytosolic concentrations and are not susceptible to artifacts derived from cell to cell variation regarding the proportion of cytosol per gram fresh weight (e.g., variable relative volume of cytosol to vacuole). ABA is a weak acid with a $pK_a$ of 4.7, and diffusion of the protonated lipophilic species from the acidic apoplasm to the alkaline cytosol had been suggested as a dominant transport mechanism (i.e., ion-trap model, **Kaiser and Hartung, 1981**; **Daie et al., 1984**). However, the finding that ABA transport into *Arabidopsis* root cells is not concentrative and relatively slow suggests that diffusion of the protonated lipophilic species is not the

dominant transport mechanism in *Arabidopsis* roots under these growth conditions. ABACUS responses in root cells indicate that cytosolic ABA concentrations are at or below the concentration of ABA applied exogenously, rather than higher than outside as expected if transport were energy-driven. The ion-trap model is also not consistent with the ABA transport properties measured in yeast cells expressing ABACUS1-2μ, since ABA applied to yeast cells did not result in a ratio change of ABACUS1-2μ expressed in the yeast cytosol, even when supplied at 20 μM (±)-ABA in a solution of pH 4.7 (*Figure 5*). An ion-trap transport mechanism would be expected to result in cytosolic ABA above the apparent $K_d$ of ABACUS1-2μ at this concentration and pH; thus the lack of ABACUS1-2μ responses indicates that yeast requires carrier-mediated uptake for importing sufficient levels of ABA for detection by ABACUS1-2μ. Several ABA transporters have recently been identified (*Boursiac et al., 2013*). Using ABACUS1, we detected concentration of ABA into yeast cells co-expressing the high affinity ABA importer AIT1/NRT1.2. AIT1/NRT1.2 and ABCG40 are high-affinity transporters that likely use active transport mechanisms and are expressed around vascular tissues in *Arabidopsis* roots (*Figure 12*), two characteristics that do not match ABA import into the cytosol of root cells expressing ABACUS1. The presence of a broadly expressed uniporter for ABA in the plasma membrane of *Arabidopsis* root cells would best explain the characteristics of ABA transport activity measured using ABACUS1.

ABA is known to impact upon its own metabolism through various feedback mechanisms. Plant cells rapidly metabolize ABA (*Daie et al., 1984*) and ABA treatment accelerates ABA catabolism (*Ren et al., 2007*). ABA treatment also induces the expression of ABA biosynthetic enzymes, potentially as part of a priming mechanism (*Barretto et al., 2011*; *Pantin et al., 2013*). In Arabidopsis roots growing in the RootChip16, ABA treatment was not required for the induction of root cell transport activity, suggesting the transporter(s) involved are constitutively expressed. However, ABA does affect its own metabolism and/or compartmentation in root cells as evidenced by a reduction in the apparent accumulation rate and increase in the apparent elimination rate of ABA in root cells that have experienced a prior treatment with ABA (*Figure 6*). This observation is consistent with the induction of *CYP707A*, which encodes the key enzyme for the first step in ABA degradation (*Kushiro et al., 2004*). Whether the acceleration by ABA of its own elimination from the cytosol is due to increased transport out of the cytosol to other compartments, increased conversion of ABA to ABA-glucose ester (e.g., through activation of ABA glycosyltransferases or inhibition of β-glucosidase 1 [BG1]), increased activity of ABA catabolic enzymes, or some combination of these changes remains to be determined.

Biosensors such as ABACUS can be targeted to sub-cellular compartments to reveal sub-cellular analyte patterning and dynamics (*Jones et al., 2013*) undetectable with current methods. For example, three different locations (apoplasm, *Baier et al., 1990*; endoplasmic reticulum, *Lee et al., 2006*; vacuole, *Xu et al., 2012*) have been proposed for β-glucosidase activity on ABA-glucose ester pools, thus ABACUS could be targeted to these compartments to elucidate sub-cellular ABA distribution. Additionally, we expect that ABACUS1[NLS] will facilitate investigation of the spatial patterning of ABA dynamics (*Figure 12*) and that future experiments performed with ABACUS sensors can shed further light on the spatiotemporal regulation of ABA transport and metabolism.

## Materials and methods

### Gateway Destination vector series

pFLIP vectors are Gateway Destination vectors carrying fluorescent proteins of a FRET pair flanking the Gateway cassette (chloramphenicol resistance marker and the *ccdB* gene between attR1 and attR2 sites). In pFLIP2 (*Kaper et al., 2008*), the *eCFP* gene, Gateway cassette, and *Venus* gene are inserted into the multiple cloning site of a pRSET-B bacterial expression vector backbone (Invitrogen, Grand Island, NY). After recombination with a ligand-binding domain from a separate Gateway Entry clone, the fusion construct contains an N-terminal eCFP, the linker encoded by the attB1 site, the ligand-binding domain, the linker encoded by the attB2 site, and a C-terminal Venus. To generate a series of pFLIPi destination vectors for expression in *E. coli* with different fluorescent protein combinations, the 5′ fluorescent protein genes were exchanged using the *Xho*I and *Kpn*I restriction sites and the C-terminal fluorescent protein genes were exchanged using the *Spe*I and *Psp*OMI sites (*Table 1*; *Supplementary file 1*). For yeast expression, a similar series of Destination vectors, termed pDR-FLIPs, was constructed by mobilizing the entire FLIP cassette from pFLIPi vectors (containing the two fluorescent proteins and the Gateway cassette) into a pDR196 vector (*Loqué et al., 2007*) modified to

contain *Xba*I and *Eag*I sites at the 5′ end of the multiple cloning site (pDR196-X). Specifically, the pFLIPi vectors were digested with *Xba*I, *Psp*OMI and the fragment containing the FLIP cassette was ligated into the *Xba*I and *Eag*I sites of the pDR196-X vector. To generate Destination vectors for plant expression under the control of the *UBQ10* promoter, termed pPZP-FLIPs, the pFLIPi vectors were digested with *Xho*I and *Apa*I and the fragment containing the expression cassette was ligated into the *Xho*I and *Apa*I compatible *Sal*I and *BstX*1 sites of the pPZP UBQ10-Kan vector. The pPZP UBQ10–Kan vector was created by replacing the CaMV 35S promoter of pPZP312 (*Hajdukiewicz et al., 1994*) with a fragment containing the *UBQ10* promoter and a kanamycin resistance cassette flanked by *Sal*I and *BstX*I sites (*Xho*I and *Apa*I compatible ends, respectively). All destination vector sequences are provided in *Supplementary file 1*.

## Gateway entry clone library

Putative ligand binding domains for ABA sensors (i.e., potential ABA sensing domains, or PAS) were selected from the ABA co-receptor complex components, specifically nine of the PYR/PYL/RCAR proteins and three of the PP2C clade A phosphatases (*Supplementary file 1*). Single domain PAS (sPAS) were designed and PCR amplified with primers containing attB sites for recombination into pDONR221 (Invitrogen) in a Gateway BP reaction (5′ primers had attB1 and 3′ primers had attB2 sites). Double domain PAS (dPAS) were constructed by fusing a PYR/PYL/RCAR to a PP2C via a variable linker region. Double domain dPAS were designed and amplified in two PCR steps. In the first step, N-terminal domain sequences were amplified with 5′ primers containing the attB1 site and 3′ primers containing a 30 base pair sequence for overlapping PCR. C-terminal domain sequences were amplified with 5′ primers containing the same 30 base pair sequence for overlapping PCR and 3′ primers containing the attB2 site. In the second step, two products from the first reactions were amplified together in an overlap extension PCR reaction and the resulting product was recombined into pDONR221 in a BP reaction. The 30 base pair overlap sequence contains *Asc*I and *Fse*I restriction sites for subsequent insertion of additional linker sequences. Primers for amplification are listed in *Supplementary file 1*. After the BP reaction, the resulting PAS Entry clones were digested with *Asc*I and *Fse*I (NEB, Ipswich, MA) and four additional linker sequences were inserted. Thus, for each unique double domain sensor construct a total of five dPAS Entry clones were generated, containing sequences coding for five distinct linkers between two ABA binding domains (*Table 1*).

## Generation of ABACUS affinity mutants

The PAS Entry clone for ABACUS1 was altered using QuikChange (Agilent, Santa Clara, CA) site-directed mutagenesis according to manufacturer's instructions to generate the ABACUS affinity mutants. All primers for site-directed mutagenesis are included in *Supplementary file 1*.

## Generation of nuclear localized ABACUS variants

To generate a pFLIPi38 destination vector for expression of a nuclear localized ABACUS, the 5′ edCitrine of pFLIPi38 was excised using the *Xba*I and *Kpn*I restriction sites and replaced with an *Xba*I and *Kpn*I digested PCR product containing a 5′ *Xba*I site followed by a *Bsp*HI site, a sequence coding for an SV40 derived nuclear localization signal (NLS, LQPKKKRKVGG; *Kalderon et al., 1984*), and edCitrine. The primers used to amplify NLS-edCitrine are provided in *Supplementary file 1*. The resulting vector was termed pFLIPiNLS38. To generate a Destination vector for plant expression of nuclear localized ABACUS under the control of the *UBQ10* promoter, the pFLIPiNLS38 vector was digested with *Bsp*HI and *Apa*I and the fragment containing the expression cassette was ligated into the *Bsp*HI and *Apa*I compatible *BstX*1 site of the pPZP UBQ10-Kan vector. All destination vector sequences are provided in *Supplementary file 1*.

## Gateway cloning of sensor expression clones

LR reactions were performed in 96 well format using 1 µl of pFLIP Destination vector (~25 ng/µl), 1 µl of PAS Entry clone (~25 ng/µl) and 0.5 µl LR clonase II (Invitrogen). Samples were incubated 1–18 hr at 25°C and then added to 50 µl of chemically competent TOP10 *E. coli* cells (Invitrogen). Bacterial transformation was performed according to manufacturer instructions (Invitrogen) except that after heat shock and dilution in SOC medium, 50 µl of the cell solution was again diluted in 800 µl Luria–Bertani (LB) medium containing 100 µg/ml carbenicillin (Sigma, St. Louis, MO) and cultured overnight. These cultures were then diluted again (20 µl in 2000 µl) in LB medium with carbenicillin and cultured

overnight. Expression plasmids were isolated in 96-well format using the NucleoSpin 96 Plasmid kit according to manufacturer's instructions (Machery-Nagel, Düren, Germany).

## Expression of sensors in protease deficient yeast

*Saccharomyces cerevisiae* strain BJ5465 (ATCC 208289 [*MATa ura3-52 trp1 leu2- Δ1 his3-Δ200 pep4::HIS3 prb1-Δ1.6R* can*1 GAL*] [*Jones, 1991*]) was transformed with yeast expression plasmids in 96-well format using a lithium acetate transformation protocol (*De Michele et al., 2013*). Transformed yeast was plated on synthetic complete (SC) medium with 0.8% agar (Difco BD, Franklin Lakes, NJ) supplemented with 240 mg/l leucine, and 20 mg/l tryptophan (SC agar +Leu,Trp) to select for complementation of uracil auxotrophy by the URA3 marker of the pDR FLIP based expression clones. Transformants were grown in 2 × 1 ml cultures in SC medium +Leu,Trp in 96-well culture blocks (Greiner, Monroe, NC) for preliminary fluorescence analysis of sensor expression and for high-throughput screening of sensors in cleared cell lysates. For metal-affinity chromatography purification of sensors, yeast strains expressing sensors were grown in 30 ml cultures in SC medium +Leu, Trp in 50 ml culture tubes.

## Other host systems for sensor expression

Although measurement of sensors in intact *E. coli* (BL21-CodonPlus-RIL) and yeast (23344C [MAT α ura3], isogenic with Σ1278b [*Bechet et al., 1970*]) cells revealed that sensors were expressed and contained both fluorophores, we observed loss of FRET and protein cleavage upon cell lysis (*Figure 2*). Therefore, expression in these host systems were not further pursued.

## Fluorescence analysis of cleared cell lysates

Yeast cell cultures (OD600 ~0.5) were centrifuged at 5000×*g* for 7 min, washed once in 1 ml PBS buffer (137 mM NaCl, 2.7 mM KCl, 10 mM Na$_2$HPO$_4$, 1.8 mMKH$_2$PO$_4$, pH7.4), transferred to a 96-well culture block in the case of 30 ml cultures, and centrifuged a second time at 5000×*g* for 7 min. The cell pellets were then frozen at −80°C. Frozen cell pellets were thawed and resuspended in 1 ml PBS buffer. A 50 µl aliquot was transferred to a clear bottom microtiter plate (Greiner) for analysis of whole cell fluorescence. Cells were then centrifuged at 5000×*g* for 7 min, supernatant was discarded and 700 µl of chilled glass bead slurry (PBS buffer, 0.1% Triton X-100 and 50% vol/vol 0.5 mm Zirconia/Silica beads [BioSpec, Bartlesville, OK]) was added to each cell pellet. Culture blocks were then sealed with aluminum foil tape (3M 439 Silver, 3-1/4 in width, 3M, Saint Paul, MN) and vortexed to resuspend cells. The block was then loaded into a MM300 mixer mill (Retsch, Haan, Germany), which was run for 15 min at a frequency setting of 20. Cell lysate was centrifuged at 5000×*g* for 10 min at 4°C and 200 µl supernatant was transferred to a microtiter plate and centrifuged again (5000×*g* for 5 min) to remove remaining cell debris. The resulting supernatant was then diluted in PBS buffer for use in fluorescence analysis in clear bottom microtiter plates using a Safire fluorimeter (Tecan, Männedorf, Germany). Samples were diluted in 20 mM MOPS pH 7.0 in order to obtain fluorescence emission at 480 nm and 530 nm between 5000 and 50,000 relative fluorescence units (RFU) with an excitation wavelength of 428 nm and a gain between 70 and 100. For all fluorescence measurements, bandwidth was set to 12 nm, number of flashes was 10, and integration time was 40 µs. Fluorescence readings were acquired for donor fluorophore (eCFP excitation 428 nm, eCFP emission 485 nm, abbreviated DxDm), acceptor fluorophore (eYFP excitation 500 nm, eYFP emission 535 nm, abbreviated AxAm) and for energy transfer from donor to acceptor (eCFP excitation 428 nm, eYFP emission 530 nm, abbreviated DxAm). Additionally, a fluorescence emission scan reading from 470 to 550 nm (step size 5 nm) with an excitation wavelength of 428 nm was acquired. Cell lysates from BJ5465 yeast containing an empty vector were used as a negative control for background subtraction. To analyze sensor response to ABA, 100 µl of sensor samples were combined with 50 µl of an ABA in water solution (diluted from a 20 mM [±]-ABA racemic mixture in 50 mM NaOH]) or a NaOH in water mock control. ABACUS1-2µ response to (+)-ABA alone was indistinguishable from (+)-ABA as one half of a (±)-ABA racemic mixture (*Figure 3F*). Replicates were averaged and a ratio of DxAm/DxDm was calculated as an approximation of FRET ratio. ABA-dependent ratio change was calculated as the FRET ratio with ABA over FRET ratio with mock treatment.

## Fluorescence analysis of metal affinity chromatography purified sensors

For high-throughput/low-yield sensor purification, sensors were purified from cleared cell lysates in 96-well format according to manufacturer's instructions using the HisPur cobalt resin system (Pierce (Thermo Scientific), Rockford, IL). For low-throughput/high-yield sensor purification, lysates were

diluted 1:2 in 50 mM Sodium phosphate, 300 mM NaCl, 5 mM Imidazole pH 7.4 and then filtered through a 0.45 µm PES filter and bound to Poly-Prep chromatography columns (Bio-Rad) containing His-bind resin (Novagen (EMD), Madison, WI). Columns were then washed twice with 50 mM Sodium phosphate, 300 mM NaCl, 5 mM Imidazole pH 7.4 and eluted in 50 mM Sodium phosphate, 300 mM NaCl, 200 mM Imidazole, pH 7.4. Samples were diluted and analyzed on a Tecan Safire fluorometer as described for cleared cell lysates. Determination of the apparent $K_d$ of the ABACUS variants was performed as described previously (*Deuschle et al., 2005a*). Testing ABACUS response to other compounds was performed as described above for ABA except that stock solutions were prepared as follows: pyrabactin was dissolved in DMSO, indole-3-acetic acid (IAA), jasmonic acid (JA), kinetin (Kin), salicylic acid (SA) and gibberellic acid A3 (GA3) were dissolved in 50% ethanol, NaCl, KCl and glucose were dissolved in water. Mock solutions contained the appropriate solvent. All chemicals were from Sigma unless otherwise noted. For low-throughput sample analysis, data are reported as means and standard deviations of 3–4 replicates and each experiment was performed at least three times with similar results.

## Analysis of transporter activity in yeast cells

Entry clones for known or potential ABA transporters (AIT1–AT1G69850, ABCG40–AT1G15520) were recombined with the pXC-DR GWY Destination vector, and the yeast strain BJ5457 (ATCC 208282 [*MATalpha ura3-52 trp1 lys2-801 leu2-Δ1 his3-Δ200 pep4::HIS3 prb1-Δ1.6R can1 GAL*, *Jones, 1991*]) was transformed with the resulting expression clones. The pXC-DR GWY vector was created with SLIC (*Li and Elledge, 2007*) by joining the 5713 bp *KpnI* and *SacII* fragment from the pMetYC GWY vector (*Lalonde et al., 2010*) with a PCR product containing the PMA1 promoter fragment and gateway cassette amplified from the pDR-GWY vector (*Loqué et al., 2007*). pXC-DR GWY carries a Leu2 gene for complementation of BJ5457 leucine auxotrophy. Expression of transporters in the BJ5457 haploid allowed for combinatorial mating with BJ5465 yeast strains expressing ABACUS1-80µ, ABACUS1-2µ, and ABACUS1-nr. Diploid yeast strains expressing transporters and sensors were selected on SC agar medium +Trp. Diploid yeast strains were cultured in SC medium +Trp overnight and washed twice in SC medium +Trp and then resuspended in 20 mM MES buffer, pH 4.7. 135 µl of yeast samples was mixed with 15 µl of 20 mM MES pH 4.7 containing 10 X (±)–ABA solution or appropriate NaOH mock solution for 5 min. Fluorescence readings were collected using a Tecan Safire fluorometer and analyzed as described above for cleared cell lysates except that a diploid yeast strain with pDR-GWY and pXC-DR GWY empty vectors was used as a control for background subtraction. Data are reported as means and standard deviations of 3–4 replicates and each experiment was performed at least three times with similar results.

## Transgenic plant lines expressing ABACUS

Transgenic plant lines were generated using the Agrobacterium floral dip method with the Col0 *rdr6-11* silencing mutant (*Peragine et al., 2004*) as described previously (*Deuschle et al., 2006*). Transformants were selected on agar plates containing ½ × MS medium with BASTA. Fluorescence expression of transformants on agar plates was analyzed using a FluorChem Q imager (Alpha Innotech, San Leandro, CA) with CY2 excitation and emission and the following settings: 12 s exposure time, normal speed, ultra resolution and level 2 noise reduction. All ABACUS plant lines are summarized in *Supplementary file 1*.

## Phenotypic characterization of plant lines expressing ABACUS

For root length assays, plant lines were grown vertically on ½ × MS agar medium (½ × MS salts [PhytoTechnology Laboratories, Shawnee Mission, KS], 0.8% Agar [BD], 2.56 mM MES pH 5.7) for 5 days and then transferred to ½ × MS agar medium plates or plates supplemented with 100 mM NaCl, 200 mM sorbitol or (±)-ABA as indicated. For ABA treatments, 20 mM (±)-ABA was prepared freshly as a stock solution in 50 mM NaOH. Mock treatment plates were made using NaOH to keep pH consistent. Root tips were marked at the time of transfer and images were acquired after 4 days growth on vertical treatment plates. Root growth after transfer were measured using FIJI (http://fiji.sc/). Each experiment included ~15 roots per genotype and was repeated three independent times with similar results. ABA sensitivity of seed germination rate of Arabidopsis seeds was assayed after sowing on ½ × MS agar medium plates supplemented with 0, 2 or 20 µM (±)-ABA. All genotypes showed 100% germination on ½ × MS agar medium plates and 0% germination on plates supplemented with 20 µM (±)-ABA. Plates were stratified for 2 days at +4°C after which they were transferred to a growth chamber with standard long day conditions (16 hr high light, 22°C, 70% RH). Seeds were considered germinated at radicle emergence and germination rate was assayed 9 days post sowing in three

independent experiments of 100–200 seeds per genotype. The *abi1-2*, *hab1-1*, *pp2ca-1* triple mutant line was a gift from Pedro Rodriguez (**Rubio et al., 2009**).

## RootChip16 device fabrication

The RootChip16 was designed using AutoCAD software (Autodesk) and fabricated essentially as described for the original RootChip (**Grossmann et al., 2011**, **2012**). In short, designs were reproduced onto emulsion photomasks and molds for flow and control layers were fabricated with SU-8 photoresist on 4-inch silicon wafers (Stanford Microfluidics Foundry) (**Anderson et al., 2000**). Height of control layer features was 20 μm; height of the flow channels and root observation chambers (both on flow layer mold) was 20 μm and 100 μm, respectively. Both chip layers were produced by pouring polydimethylsiloxane and RTV-615 onto chlorotrimethylsilane-treated molds. The control layer was spin-coated to a height of 30 μm. Both layers were baked 1 hr at 80°C, pealed off the wafers, trimmed to size, and holes were punched with 20 gauge (0.812 mm) for solution inlets (flow layer), 14 gauge (1.628 mm) for solution outlets. Root inlets were punched in an angle of 30° to the normal with 18 gauge (1.024 mm). The assembled device was postbaked over night at 80°C and additional 20-gauge holes were punched for the control lines (control layer). Completed chips were plasma bonded to optical glass (No.1, thickness 0.15 mm) and baked for 5 min at 80°C.

The chip carrier was designed using AutoCAD and printed on a ProJet 3500HD (3D Systems, Rock Hill, SC) with a central aperture for the chip (42 × 60 mm). Outer dimensions were 110 × 160 mm to fit into the insert aperture of a motorized microscope stage (ASI, Carlsbad, CA). The carrier further contained water reservoirs to maintain high humidity during plant growth at the microscope. Mask designs and fabrication protocols will be made available upon request.

## Seedling growth, RootChip16 device preparation, and root perfusion

The RootChip16 device was used as described in detail previously for the RootChip (**Grossmann et al., 2011**, **2012**). Briefly, the device was sterilized by UV light exposure, immersed in liquid growth media (¼ × MS salts, 1.28 mM MES pH 5.7 supplemented with B vitamins at final concentrations of niacin 2.3 μM, thiamine 1.5 μM, pyridoxin 1.2 μM and myo-inositol 0.3 μM), and air in the root observation chambers was removed by applying suction at the solution outlets using a microliter pipette. 5-day old Arabidopsis seedlings, germinated on 5 mm long cut 10 μl pipette tips filled with solidified growth medium (½ × MS salts, 1% Agar, 2.56 mM MES pH 5.7 supplemented with B vitamins at final concentrations of niacin 4.6 μM, thiamine 3.0 μM, pyridoxin 2.4 μM and myo-inositol 0.6 μM, were transferred onto the chip by plugging the tips into the root inlets. The chip was incubated for 24 hr under standard long-day growth conditions (16 hr high light, 22°C, 70% RH) to allow roots to grow into the observation chambers. Before imaging, the RootChip16 was inserted from below into the central aperture of the carrier and fixed with tape. Pressure lines for control over micromechanical valves and flow lines for perfusion of solutions were connected. To actuate the push-up valves on the chip, a closing pressure of 15 p.s.i (1.03 bar) was applied to the water filled control lines. Solution vials were pressurized with 5–10 p.s.i. (0.34 bar) to drive perfusion solutions through the flow lines of the device.

## *In planta* analysis of ABACUS using the RootChip16

The RootChip 16 was perfused with low strength growth medium (¼ × MS salts, 1.28 mM MES pH 5.7). For experiments, 10 mM (+)-ABA was prepared freshly as a 20 mM (±)-ABA stock solution in 50 mM NaOH and diluted in low strength growth medium. Mock solutions for the different experiments were made using NaOH as to keep pH constant. Vials with solutions were connected to root chip flow lines and pressurized. Perfusion of solutions was controlled by the RootChip 16 control valves (**Grossmann et al., 2011**, **2012**) connected to a valve controller controlled by a LabView (National Instruments) program.

## *In planta* analysis of ABACUS responses in detached leaves

Arabidopsis plants were grown on ½ × MS agar medium (½ × MS salts, 0.8% Agar, 2.56 mM MES pH 5.7) for 5 days and then transferred to soil and grown in a growth chamber (16 hr light, 22/18°C, 50% RH). Leaves 5 and 6 were harvested at 21–25 days post sowing and their petioles were placed in solution containing ¼ × MS medium (¼ × MS salts, 1.28 mM MES pH 5.7) or the same medium supplemented with 150 mM NaCl or 100 μM (±)-ABA. Leaves were incubated in a growth chamber (16 hr high light, 22°C, 70% RH) for 24 hr and then imaged (60 μm Z-stacks were acquired for each position imaged). Data are presented as means and standard deviations of the average ratio value for five leaves with three images/leaf.

## Fluorescence microscopy

RootChip and detached leaf imaging was performed using either a 5x 0.12 N PLAN, 10x 0.40 HC PL APO, or 20x 0.70 HC PL APO objective (Leica, Wetzlar, Germany) on an inverted epifluorescence microscope (Leica DM IRE2), equipped with a motorized stage (Scan IM 127x83; Märzhauser Wetzlar, Wetzlar, Germany), a Polychrome V monochromator light source (TILL Photonics, München, Germany)), and an electron multiplying charge-coupled device camera (QuantEM:512SC; Photometrics, Tucson, AZ). A DualView beam splitter (Photometrics) containing an ET470/24m and ET535/30m filter setup allowed simultaneous imaging of donor and acceptor emissions for FRET measurements as well as acceptor excitation/emission for normalization of sensor expression levels or for detection of artifacts that may affect the acceptor fluorescent protein. Imaging data were acquired using SlideBook 5.0 software (Intelligent Imaging Innovations, Denver, CO). Data were acquired as multi-position time series with simultaneous acquisition of FRET donor and acceptor emission under donor excitation, followed by acquisition of acceptor emission under acceptor excitation. A typical acquisition used an intensification setting of 300, a gain setting of 3, binning of $2 \times 2$, 400 ms exposures for DxDm and DxAm, 100 ms exposure for AxAm, and imaging intervals of 1 min.

Confocal images were acquired on a Leica SP5 using a 20x 0.70 HC PLAN APO objective. 442 nm and 514 nm lasers were used for excitation of donor and acceptor, respectively. Fluorescence emission was detected by PMT detectors, set to detect 458–482 nm for donor emission and 520 to 550 nm for acceptor emission.

## Image processing and analysis

Image processing and analysis were performed using FIJI (http://fiji.sc/) and Matlab (MathWorks). To correct for movement of regions of interest due to root growth, images were registered using the MultiStackReg v1.4 plug-in (*Thevenaz et al., 1998*, modified by Brad Busse). When needed, multi-point time series were stitched together using the grid/collection stitching plug-in (*Preibisch et al., 2009*) using a maximum intensity fusion method. Mean intensity values for regions of interest were calculated as follows: Background was subtracted from all measured intensities; donor (DxDm) and acceptor (DxAm) intensities under donor excitation were corrected against acceptor intensity under acceptor excitation (AxAm) to correct for intensity fluctuation caused by focus drift, root movement, or changing sensor protein levels (during long-term measurements). Ratios of DxAm/DxDm were calculated and data were normalized to time point zero. For detached leaf imaging, 60 µm Z-stacks of DxAm and DxDm images were maximum projected and then background subtracted. Background was calculated from average DxAm and DxDm values for *rdr6-11* leaves (n = 20). Average ratios of DxAm/DxDm were calculated for each image. Data are presented as means and standard deviations of five leaves with three images per leaf. For calculating ratio images of the confocal aquisitions of ABACUS1-2µ, z-stacks of the DxDm and DxAm channels were summed before taking the ratio of the summed DxAm and summed DxDm.

## Acknowledgements

We thank Heather Cartwright for help and advice with imaging and image analyses and Clare Gill for excellent technical assistance. Yeast strain 23344C was kindly provided by Bruno André, ULB, Charleroi, Gosselies. We thank Emre Araci and the Stanford Microfluidics Foundry for advice on mold design and fabrication of RootChips, and Rafael Gómez-Sjöberg (LBNL) for providing details about the construction of the valve controller.

## Additional information

### Funding

| Funder | Grant reference number | Author |
| --- | --- | --- |
| National Science Foundation | EAGER IOS-1045185 | Wolf B Frommer |
| Swedish Research Council | | Jonas ÅH Danielson |

The funders had no role in study design, data collection and interpretation, or the decision to submit the work for publication.

## Author contributions

AMJ, Conception and design, Acquisition of data, Analysis and interpretation of data, Drafting or revising the article; JÅHD, performed experiments, imaging, analyzed data, revised manuscript, prepared figures, Acquisition of data, Analysis and interpretation of data, Drafting or revising the article; SNMK, Acquisition of data; VL, Acquisition of data, Analysis and interpretation of data, Drafting or revising the article; GG, Drafting or revising the article, Contributed unpublished essential data or reagents; WBF, Conceived of concept, supervised overall project and participants, analyzed data and wrote and revised manuscript, Conception and design, Analysis and interpretation of data, Drafting or revising the article

## Additional files

### Supplementary files

• Supplementary file 1. Details for pFLIP Destination vectors, PAS Entry clones, L linkers, primers, and ABACUS plant lines.

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
