## [Decision Letter]

Thank you for sending your work entitled “Abscisic acid dynamics in roots detected with genetically encoded FRET sensors” for consideration at *eLife*. Your article has been favorably evaluated by a Senior editor, Detlef Weigel, and 3 reviewers, one of whom, Richard Amasino, is a member of our Board of Reviewing Editors.

Two of the issues require further experimentation.

1) Determine whether transgenic lines expressing the sensors are able to report endogenous changes in ABA concentration. This experiment is not overly burdensome as the transgenic lines are already in place and as a reviewer notes “mannitol treatment of seedlings would trigger endogenous ABA synthesis and could be used a fast/simple experiment to prove that the sensors are useful for measuring changes in endogenous ABA content.”

2) Determine whether transgenic lines expressing the sensors exhibit altered sensitivity to exogenous ABA. Because the sensors bind to ABA and possibly to other endogenous proteins required for ABA signal transduction, there is the issue of whether or not plants expressing the sensors exhibit altered ABA sensitivity. Again, this experiment is not overly burdensome; for example, straightforward growth inhibition assays could be used. As a reviewer notes “I don't think that the lines need to have 100 % wild type ABA sensitivity to be useful. It would be ideal if this is the case, but the key point is to know their inherent properties and limitations and the caveats that come along with using them.”

Other issues:

1) The authors use the *rdr6*-11 gene silencing mutant of *Arabidopsis* to overcome limitations on silencing of their FRET sensors in vivo. Some discussion of whether there are or aren't any reports of ABA-related phenotypes in this background would be useful.

2) In Figure 6 why does the ABACUS1-nr decline with repeated ABA challenges?

3) There is an older literature focused on direct measurements of ABA in defined cells after stress. For example Weiler's group (48) suggested that the concentration of ABA in guard cells (and other cells) after drought stress was on the order of 10 μM. The Harris et al. paper cites examples of other work in the area. I was a little surprised that the authors did not cite this older work...”

4) The experimental concentrations of exogenously supplied ABA that are reported are specifically with respect to the concentration of the naturally occurring (+)-stereoisomer; however, mixed stereoisomers were used in all of the experiments. The (-)-stereoisomer is biologically active and has activity on the ABA receptors, albeit substantially reduced. Competition between stereoisomers could potentially affect rates of transport. These issues make interpreting the data more complicated than if the pure (+)-stereoisomer was used (which is readily available and relatively inexpensive). This complicating issue should be explicitly addressed in the manuscript.

---

## [Author Response]

Thanks for the excellent and constructive review. We have addressed the two key comments experimentally as detailed below: whether the sensors report endogenous changes in ABA and whether the presence of the sensors may affect the physiology of the plants. In the meantime we also obtained plants that express a nuclear-localized version of ABACUS1-2µ and now provide confocal images demonstrating nuclear localization and show that nuclear-targeted sensors respond similar to the cytosolic sensors to external ABA addition, allowing for improved analysis of cell specific responses.

*1) Determine whether transgenic lines expressing the sensors are able to report endogenous changes in ABA concentration. This experiment is not overly burdensome as the transgenic lines are already in place and as a reviewer notes “mannitol treatment of seedlings would trigger endogenous ABA synthesis and could be used a fast/simple experiment to prove that the sensors are useful for measuring*
*changes in endogenous ABA content.”*

We had, over the past year, performed a large number of experiments trying to test whether, for example, salt treatment would affect cytosolic ABA accumulation. We had not included these experiments, since the results were negative, i.e., even over longer time periods, we did not observe reliable FRET sensor responses that are compatible with an increase of ABA in response to salt treatment of roots. We would like to note that we are measuring cytosolic ABA levels due to cytosolic localization of the ABA sensor, thus are not able to see accumulation of ABA in the vacuole. One possible explanation could be that cytosolic levels do not change dramatically. Our sensors report changes in the linear detection range around the Kd, thus it is also conceivable that the cytosolic accumulation is at or below the lower end of the detection range (< 400-500 nM). We also would like to note that our prototype sensors appear to respond with changes in brightness to ABA treatment of roots, effects that affect the quantitative analysis over extended time periods. Thus, for long time courses it will be necessary to reexamine the salt response using a higher affinity sensor as well as sensors that are not changing brightness in response to ABA. We will, in a next step, analyze the other ABA sensors identified in the screen in more detail to identify higher affinity sensors as well as compare the response of sensors that use different PYR/PYL/RCAR polypeptides as part of the sensory domain (see Figure 1—figure supplement 1).

We performed a large number of additional experiments using salt treatment of leaves (feeding via the petiole). These experiments are now included as Figure 10. The responses were variable between experiments, however ABACUS1-2µ showed statistically significant responses in ABA treatment, and a clear trend of increased ABA accumulation in response to salt treatment.

*2) Determine whether transgenic lines expressing the sensors exhibit altered sensitivity to exogenous ABA. Because the sensors bind to ABA and possibly to other endogenous proteins required for ABA signal transduction, there is the issue of whether or not plants expressing the sensors exhibit altered ABA sensitivity. Again, this experiment is not overly burdensome; for example, straightforward growth inhibition assays could be used. As a reviewer notes “I don't think that the lines need to have 100 % wild type ABA sensitivity to be useful. It would be ideal if this is the case, but the key point is to know their inherent properties and limitations and the caveats that*
*come along with using them.”*

The concern of the reviewers regarding effects of the sensors on physiology are important and a point well taken. One can envision either scavenging of ABA by the sensors, thus affecting pool sizes, or the expression of the sensory domains affecting endogenous ABA signaling. To address this we expanded on our previous phenotyping experiments to now also include;

A. Root Growth assays in response to different physiologically relevant stresses (Figure 7)

B. Germination assay at different levels of ABA (Figure 7)

C. Root Growth assay at different concentrations of ABA (Figure 7)

While the root growth under stress conditions did not appear different from wild type, there was a clear hypersensitivity to ABA observed that correlated with the affinity of the sensors (both germination and root growth assays on ABA). This phenotype is similar to that found for PYR/PYL/RCAR overexpressors, indicating that despite the fusion of the truncated ABI1 domain to PYL1, our sensor can still effectively interact with endogenous PP2CAs and affect ABA signaling. Thus, similar as for Tsiens’s original calcium sensor, where the calmodulin-domain can still interact with endogenous calmodulin-binding proteins, it will be necessary to reduce these interaction with endogenous factors either by using larger PP2CA domains that have a higher affinity relative to the ABI1aid (i.e., to promote intramolecular interactions rather than intermolecular interaction) or by mutagenesis (e.g., of PYL1 S112) to block interaction with endogenous PP2CAs as successfully done for Tsien’s calcium sensor. It is noteworthy that the original calcium sensor was published in 1996, the mutants that were blocked regarding interactions with endogenous calmodulin binding proteins were published in 2004, the first use of FRET calcium sensors, and even in 2013 and 2014, calcium sensors are still being optimized for in vivo application. We would thus argue that we here not only developed the first ABA sensors, which are already significantly optimized over sensors for other solutes, but that we can use them for in vivo imaging (e.g., high-resolution imaging of cytosolic ABA accumulation and elimination rates). However, we are aware that further optimization is necessary, and that the suite of sensors already developed in our high throughput screen likely already contains sensors that can be used that address the potential drawbacks detected with this set of first generation ABA sensors.

*Other*
*issues:*

*1) The authors use the rdr6-11 gene silencing mutant of Arabidopsis to overcome limitations on silencing of their FRET sensors* in vivo*. Some discussion of whether there are or aren't any reports of ABA-related phenotypes in this background would be useful*.

To our knowledge no ABA-related phenotypes of the *rdr6-11* mutant have been reported, but since this is a very relevant concern we included both the *rdr6-11* and wild type Arabidopsis (Col0) in our phenotyping experiments (Figure 7). Thus, we were able to confirm that at least under these experimental conditions, there is no significant difference in ABA-related responses between *rdr6-11* and Col0.

*2) In*
Figure 6
*why does the ABACUS1-nr decline with repeated*
*ABA challenges?*

This is a pattern that we have observed repeatedly over longer time courses and we suspect that it is in fact a technical artifact resulting from a combination of the brightness change induced by ABA treatment and the fact that this ABACUS1-nr line has lower overall expression compared to the ABACUS1-2µ and ABACUS1-80µ line used in these experiments.

*3) There is an older literature focused on direct measurements of ABA in defined cells after stress. For example Weiler's group (*[48]*) suggested that the concentration of ABA in guard cells (and other cells) after drought stress was on the order of 10 μM. The Harris et al. paper cites examples of other work in the area. I was a little*
*surprised that the authors did not cite this older work…”*

We were not aware of this important work and have added the suggested reference and discussion to the text (Results section entitled “ABA accumulation in leaves”): “Dissection of water-stressed *Vicia faba* leaves and quantitative ABA analyses indicated that ABA accumulates in epidermis, mesophyll and guard cells to ∼10 µM when ignoring compartmentation (PMID: 16593922).”

*4) The experimental concentrations of exogenously supplied ABA that are reported are specifically with respect to the concentration of the naturally occurring (+)-stereoisomer; however, mixed stereoisomers were used in all of the experiments. The (-)-stereoisomer is biologically active and has activity on the ABA receptors, albeit substantially reduced. Competition between stereoisomers could potentially affect rates of transport. These issues make interpreting the data more complicated than if the pure (+)-stereoisomer was used (which is readily available and relatively inexpensive). This complicating issue should be explicitly addressed in the manuscript*.

The use of (±)-ABA vs (+)-ABA has been clarified throughout the text and figures and the potential confounding effects on observed transport rates and saturation arising from use of (±)-ABA is now specifically addressed. We explicitly show now in the figures the compound added (i.e., racemic mix) and state in the figure legends that, since the sensor is specific for (+)-ABA, the actual concentration of (+)-ABA corresponds to half of the value provided for (±)-ABA.